# Inactivation of branched-chain amino acid uptake halts *Staphylococcus aureus* growth and induces bacterial quiescence within macrophages

Adriana Moldovan[1], Ronald S. Flannagan[2], Marcel Rühling[1], Kathrin Stelzner[1], Clara Hans[1], Kerstin Paprotka[1], Tobias C. Kunz[1¤], David E. Heinrichs[2], Thomas Rudel[1*], Martin J. Fraunholz [1*]

1 Department of Microbiology, Biocenter, University of Würzburg, Würzburg, Germany, 2 Department of Microbiology and Immunology, University of Western Ontario, London, Ontario, Canada

¤ Current address: Lonza AG, Visp, Switzerland
* martin.fraunholz@uni-wuerzburg.de (MJF); thomas.rudel@uni-wuerzburg.de (TR)

## Abstract

*Staphylococcus aureus* is a notorious human pathogen that thrives in macrophages. It resides in mature phagolysosomes, where a subset of the bacteria eventually begin to proliferate. How *S. aureus* acquires essential nutrients, such as amino acids, for growth in this niche is poorly understood. Using a long-term primary human macrophage infection model, we show that branched-chain amino acid (BCAA) uptake mediated by the major transporter BrnQ1 is required by *S. aureus* for intracellular replication in macrophages and we provide mechanistic insight into the role of BCAAs in the success of intracellular *S. aureus.* Loss of BrnQ1 function renders intracellular *S. aureus* non-replicative and non-cytotoxic. The defective intracellular growth of *S. aureus brnQ1* mutants can be rescued by supplementation with BCAAs or by overexpression of the BCAA transporters BrnQ1 or BcaP. Inactivation of the CodY repressor rescues the ability of *S. aureus brnQ1* mutants to proliferate intracellularly independent of endogenous BCAA synthesis but dependent on BcaP expression. Non-replicating *brnQ1* mutants in primary human macrophages become metabolically quiescent and display aberrant gene expression marked by failure to respond to intraphagosomal iron starvation. The bacteria remain, however, viable for an inordinate length of time. This dormant, yet viable bacterial state is distinct from classical persisters and small colony variants.

## Author summary

*Staphylococcus aureus* is a prominent human pathogen causing acute and chronic disease. It is facultatively intracellular and can reside within many host cell types, including professional phagocytes such as macrophages. The

**Data availability statement:** All relevant data are within the manuscript and its Supporting Information files. RNAseq have been deposited in NCBI's Gene Expression Omnibus under GEO Series accession number GSE235929 (https://www.ncbi.nlm.nih.gov/geo/query/acc.cgi?acc=GSE235929).

**Funding:** This work was supported by the Deutsche Forschungsgemeinschaft (German Research Foundation DFG, https://www.dfg.de; RTG 2157 to T.R., RTG 2581 to M.F., CRC1583 to M.F.; DFG RU 631/17-1 to T.R.) and the Canadian Institutes of Health Research (https://cihr-irsc.gc.ca; PJT-183848 to D.E.H.). A. M. was partially supported by a grant of the German Excellence Initiative to the Graduate School of Life Sciences (GSLS; https://www.graduateschools.uni-wuerzburg.de/life-sciences/), University of Würzburg and the DAAD STIBET Fellowship of the University of Würzburg. M.R. received funds of the Bavarian State Ministry of Science and Arts and the University of Würzburg to the Graduate School of Life Sciences (GSLS), University of Würzburg. The funders had no role in study design, data collection and analysis, decision to publish, or preparation of the manuscript.

**Competing interests:** The authors have declared that no competing interests exist.

intracellular state contributes to dissemination, recurrence and infection chronicity. Chronic and relapsing infections are often associated with so-called *persister* phenotypes. Growth arrest and metabolic quiescence, accompanied by antibiotic tolerance, are hallmarks of *persistence* in bacteria. Antibiotic pressure is a major factor in triggering intracellular *persistence*. The small colony variant (SCV), an extensively studied form of *S. aureus persister*, can arise in the absence of antibiotic pressure and exhibits very distinctive phenotypic characteristics.

Here, we describe a different growth-arrested state of *S. aureus*, which conforms to the definition of a non-antibiotic-driven form of intracellular *dormancy,* triggered by branched-chain amino acid starvation in macrophages. We show that loss of function of the major branched-chain amino acid transporter BrnQ1 renders intracellular *S. aureus* non-replicative and metabolically quiescent for an inordinate period of time. Upon stochastic exit from infected macrophages, *brnQ1* mutants retain full virulence. This *dormancy* differs from classical *persistence* or SCVs and uncovers an underestimated role for BCAA uptake in the success of intracellular *S. aureus.*

## Introduction

*Staphylococcus aureus* is a Gram-positive prominent human opportunistic pathogen [1], capable of causing a variety of diseases ranging from superficial skin infections to life-threatening conditions (reviewed in [2]). It permanently colonizes 20–30% of the human population [3,4], with up to 60% carrying *S. aureus* intermittently during their lifetime [5].

S. aureus survives and replicates in a variety of host cell types including non-professional phagocytes (such as epithelial and endothelial cells) and professional phagocytes such as macrophages (reviewed by [6,7]). Recently it has been proposed that the ability of *S. aureus* to exist intracellularly is a prevailing feature of the *S. aureus*-host interaction [8]. Among professional phagocytes, macrophages play a central role in innate immunity due to their ability to phagocytose bacteria and mount potent microbicidal responses. Indeed, macrophages deploy a multitude of antibacterial effectors such as reactive oxygen species, proteases, lipases, and antimicrobial peptides that function to intoxicate and/or degrade phagocytosed prey [9]. At the same time, macrophages curtail bacterial growth by extruding essential nutrients such as iron and manganese from the phagosome lumen via the host transporter NRAMP1, which is just one aspect of host nutritional immunity [10]. In addition to metals, macrophage phagolysosomes are presumably depleted of several amino acids owing to the activity of host amino acid transporters (e.g., SLC38A9, SLC66A1/PQLC2, SLC36A1, and SLC66A4/cystinosin) localised to phagosomal or lysosomal membranes, where they extrude amino acids including leucine, neutral amino acids, cationic amino acids and cysteine, respectively, into the host cell cytosol [11]. Together, metal and amino acid transport should deplete the phagolysosome

of nutrients thereby hindering bacterial proliferation within macrophages. Despite this, macrophages fail to eradicate phagocytosed *S. aureus* [12–15] and following engulfment, bacteria can begin to replicate within mature phagolysosomes [16,17] independently of toxin production [18].

Branched-chain amino acids (BCAAs) - isoleucine (Ile), leucine (Leu) and valine (Val) - are essential amino acids that cannot be synthesized by mammals. In *S. aureus*, BCAAs are not only proteinogenic amino acids, but also serve as precursors for branched-chain fatty acids (BCFAs), which are required for the regulation of membrane fluidity [19]. In addition, Ile functions as a signalling molecule that, together with GTP [20,21], modulates the activation of CodY, a global transcriptional regulator of metabolism and pathogenesis in most low G + C Gram-positive bacteria including *S. aureus* [22].

BCAA transport and biosynthesis are therefore closely linked to virulence in *S. aureus* and other bacterial pathogens, indicating that access to BCAAs as a critical nutrient is required by pathogens to establish infection [23]. For example, when grown *in vitro* in the absence of Val, *S. aureus* displays a growth lag of ~20h, until BCAA biosynthesis is initiated [21]. To facilitate immediate BCAA uptake from the environment, *S. aureus* employs the BCAA transporters, BrnQ1 and BcaP, as well as BrnQ2, a dedicated Ile transporter [24]. In addition, *S. aureus* harbours BCAA biosynthetic genes encoded by the *ilvDBNC-leuABCD-ilvA* operon and *ilvE*. Both BCAA uptake and biosynthesis operons are repressed by CodY when intracellular BCAA levels are sufficient to support growth [21]. CodY controls the expression of over 200 genes in *S. aureus* [25–27] where it acts primarily as a repressor of metabolic and virulence genes upon binding of Ile and GTP. Upon nutrient depletion, CodY target genes were shown to be de-repressed sequentially, along the BCAA gradient, with genes involved in amino acid uptake (including BCAAs) being turned on much earlier than biosynthesis genes [28]. The requirement for BCAA acquisition for *S. aureus* pathogenesis is underlined by the fact that *brnQ1/bca*P mutants are attenuated in murine models of infection [24,29].

We here reveal a critical role for high-affinity BCAA transport in the growth and adaptation of *S. aureus* to the macrophage phagolysosome. Using defined mutants lacking specific BCAA transporters or the ability to synthesize BCAA, we uncover a pivotal role for the high-affinity BCAA importer BrnQ1 in the replication and subsequent survival of *S. aureus* within macrophages. We provide evidence that *brnQ1*- deficient *S. aureus* display a phagolysosomal growth defect that can be overcome by inactivation of the transcription factor CodY, by exogenous supply of BCAAs, or by heterologous expression of either BrnQ1 or the secondary BCAA transporter BcaP. In the absence of BCAA supplementation of the macrophage culture medium, *brnQ1* bacteria remain intracellular for an inordinate duration and adopt a *dormant* state, that is distinct from established persister cell or small colony variant (SCV) phenotypes.

## Results

### S. aureus deficient for the BrnQ1 branched-chain amino acid transporter do not replicate in human macrophages

*S. aureus* are internalized by professional phagocytes such as macrophages, where they survive and replicate within acidic phagolysosomes [18]. To test the possible involvement of BCAAs in the *S. aureus* – macrophage interaction we established a macrophage infection model, involving a short phagocytosis time, followed by a lysostaphin pulse-treatment to eradicate extracellular bacteria (Fig 1a). Here we purposefully avoided prolonged use of lysostaphin or gentamicin as antimicrobials as these can be internalised by macrophages and thus could impact the survival of intracellular bacteria [12,31–33]. Using macrophage colony-stimulating factor (M-CSF)-differentiated human monocyte-derived macrophages, we analysed the ability of individual *S. aureus* USA300 mutants retrieved from the Nebraska Transposon Library [34] to replicate and egress from infected macrophages. We analysed mutants deficient for the BCAA transporters BrnQ1, BrnQ2, BrnQ3 and BcaP, as well as mutants lacking the *ilvE*, *ilvD* and *leuA* genes, which are involved in BCAA biosynthesis [21]. A mutant in *codY*, the repressor of the aforementioned genes [26,27], known to exhibit a hypervirulent phenotype [35], was included as control (Fig 1b). This analysis revealed that by 24h post-infection (*p.i.*) all strains except the *brnQ1* mutant, had emerged from infected macrophages and massively replicated extracellularly, resulting in host cell death and the formation of micro-colonies within the infected wells of cell culture plates (Fig 1c). Since the high number of bacteria

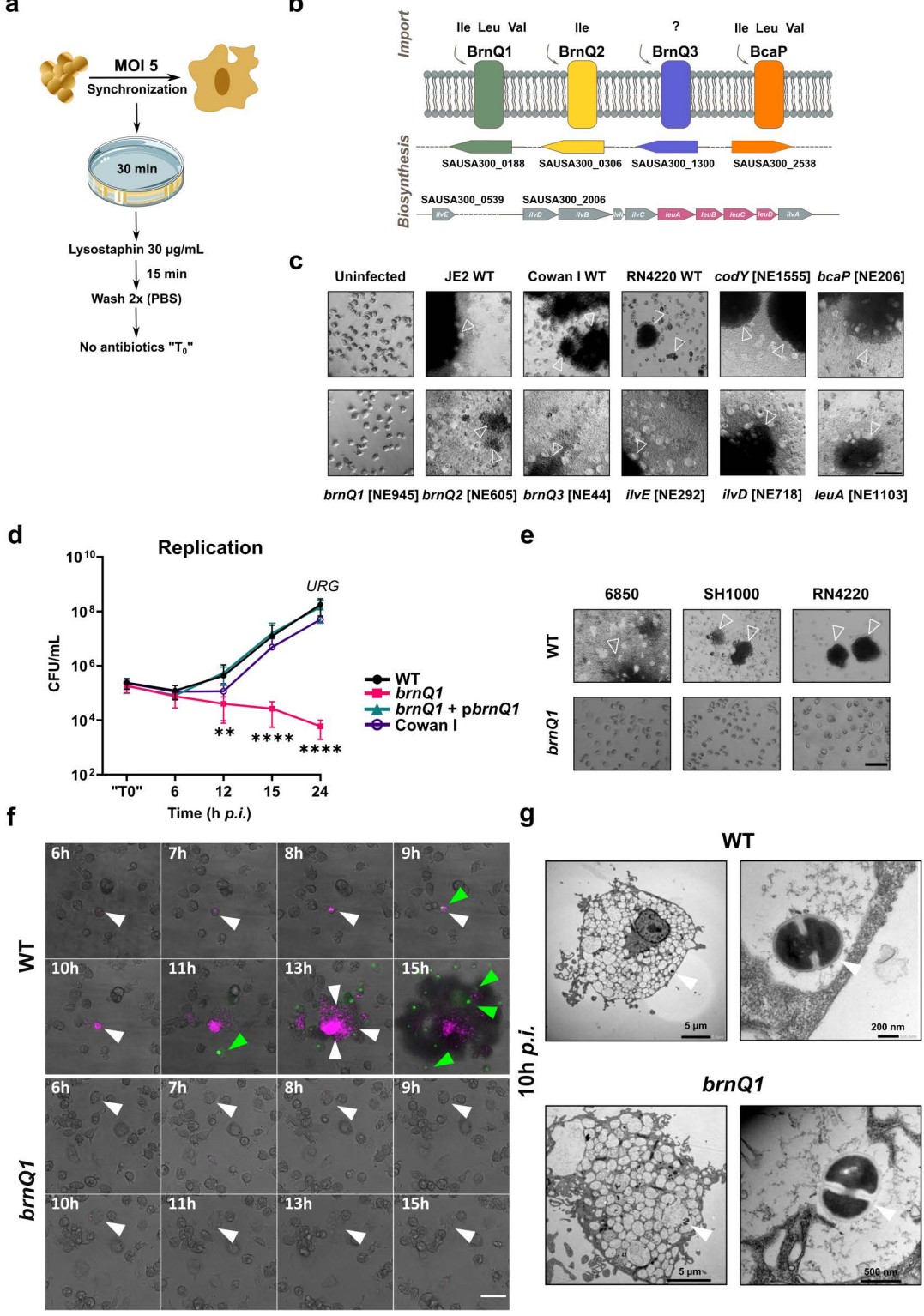

**Fig 1. The BCAA transporter BrnQ1 is required for *S. aureus* replication in human macrophages. (a)** Experimental procedure for macrophage infection: primary human macrophages were infected with *S. aureus*, at a multiplicity of infection (MOI) of 5 for 30 min. Extracellular *S. aureus* were removed by a 15 min treatment with 30 μg/ml lysostaphin and two subsequent rinses with DPBS. Infected macrophages were further incubated in

antibiotic-free media. "T0″ corresponds to 45 min *p.i.* **(b)** *S. aureus* mutants in genes involved in either BCAA-uptake (*brnQ1-3* and *bcaP*) or biosynthesis (*ilvE*, *ilvD*, *leuA*) were chosen for the analysis. Gene identifiers correspond to USA300_FPR3757(old locus ID) [30]. **(c)** Phase-contrast micrographs of macrophages at 24h *p.i.,* which stayed uninfected, were infected with wild-type *S. aureus* strains JE2, Cowan I, and RN4220, or mutants in the indicated genes. All mutants were retrieved from the Nebraska Transposon Mutant Library (NTML) generated in the USA300 JE2 background (USA300 LAC-derivative). Unrestricted growth (*URG*) of the bacteria is observed by formation of bacterial micro-colonies (white arrows). Shown are representative images from two independent experiments. Scale bar = 100 μm. "NE" designates mutant strain identifier in the NTML. **(d)** Intracellular replication in human macrophages. Data are shown as mean ±SD from independent experiments (JE2 WT and *brnQ1*: n = 7, *brnQ1* + p*brnQ1* and Cowan I: n = 3). **(e)** Macrophage infection with *brnQ1* mutants in the genomic backgrounds of *S. aureus* strains 6850, SH100 and RN4220. White arrows indicate bacterial microcolonies. Shown are representative images from three independent experiments (Scale bar: 100 μm). **(f)** Live cell imaging of macrophages infected with mRFP-expressing JE2 WT or JE2 *brnQ1* (magenta, white arrows). Host cell death (green fluorescence, green arrows) is detected by CellEvent Caspase-3/7 Green Detection Reagent. Shown are micrographs extracted from time-lapse series representative of 3 independent experiments (n = 3). Scale bar = 50 μm. (see also S1 Movie). **(g)** Transmission electron micrographs of intracellular JE2 WT or JE2 *brnQ1* confined to the phagosomal compartment (white arrows) within the macrophage, at 10h *p.i.* Shown are representative micrographs, from one experiment. Statistical analysis: **(d)** two-way ANOVA with Dunnett's multiple comparisons test (each sample vs JE2 WT, at each time-point), using log10-transformed data; \*\**p < 0.01;* \*\*\**p < 0.001;* \*\*\*\**p < 0.0001. (URG = unrestricted growth).* **(a)** contains icons which were modified (colour and design) from https://bioicons.com/, as follows: petri-dish-lid-yellow icon by Servier https://smart.servier.com/ is licensed under CC-BY 3.0 Unported https://creativecommons.org/licenses/by/3.0/.

recovered at 24h *p.i.* is not solely due to intracellular replication, but rather to extracellular replication of those bacteria that escaped macrophage restriction, we termed this phenomenon *unrestricted growth* (*URG*).

Remarkably, even *S. aureus* strains with reduced virulence, Cowan I and RN4220 [36], emerged from infected macrophages and exhibited *URG* (Fig 1c).

To further investigate the contribution of *brnQ1* to *S. aureus* growth within macrophages, we transduced the *brnQ1*::Tn transposon insertion (NE945) into a parental *S. aureus* JE2 strain, hereafter referred to as JE2 *brnQ1* and again assessed the ability of this strain to infect macrophages and display *URG*. These data revealed that, in contrast to the parental JE2 strain, JE2 *brnQ1* failed to replicate. Genetic complementation of *brnQ1 in trans* (JE2 *brnQ1* + p*brnQ1*) (S1 Fig) rescued the ability of JE2 *brnQ1* to egress from macrophages (Fig 1d). While at 6h *p.i.* bacterial loads are comparable among all strains under scrutiny (S2 Fig), at 12h *p.i.* we recovered approximately 2-fold more bacteria compared with "T0" (after lysostaphin pulse, i.e. at 45 min *p.i.*). Both parental JE2 and JE2 *brnQ1* + p*brnQ1* as well as *S. aureus* Cowan I reached *URG* by 24h *p.i.* In contrast, the number of recovered colony-forming units (CFU) for JE2 *brnQ1* dropped to ~4% of the initial infection inoculum by 24h *p.i.* (Fig 1d). Furthermore, absence of *brnQ1* abolished *URG* in several *S. aureus* strains: 6850, SH1000 and RN4220 (Fig 1e). Taken together, these data reveal that a functional BrnQ1 transporter is a general requirement for *S. aureus* growth inside macrophages, and not simply a JE2-specific phenotype.

To investigate whether the JE2 *brnQ1* growth defect within macrophages is specific to the intracellular environment, we next analysed bacterial growth in the medium used for macrophage infection assays. In an undefined medium such as TSB, which contains an abundance of amino acids and peptides, growth of JE2 *brnQ1* is indistinguishable from JE2 WT and the complemented strain JE2 *brnQ1* + p*brnQ1* (S3a Fig). In contrast, in a chemically defined medium (CDM) containing 1 mM BCAA, JE2 *brnQ1* displayed delayed growth when compared to controls (S1 Fig). In RPMI1640, where the concentrations of Val, Leu and Ile are ~ 117 μM (Val) and 131 μM (Leu and Ile, each), JE2 *brnQ1* also displays a growth delay in comparison to JE2 WT and JE2 *brnQ1* + p*brnQ1* (S3b Fig). This growth delay is still evident in *Infection Medium* (i.e., RPMI1640 + 10% v/v FBS), indicating that even in the presence of FBS the bacterium's nutritional requirement for BCAAs is not fully met (S3c Fig). Importantly, in all culture media where JE2 *brnQ1* initially displays delayed growth, the mutant eventually achieves the same bacterial density as parental JE2 (S3b and S3c Fig). Taken together, these data show that *brnQ1*-deficient *S. aureus* displays delayed growth in media with low BCAA availability (≤ 1 mM), however, in macrophages, growth of the mutant is completely halted.

Replication of *S. aureus* within macrophages precedes cell death of infected phagocytes [12]. We therefore performed live cell imaging to monitor the intracellular replication of JE2 WT and JE2 *brnQ1* in the presence of a fluorogenic caspase

3/7 substrate (Fig 1f and S1 Movie). JE2 WT commenced replicating within macrophages at approximately 8h *p.i.*, which was followed by bacterial egress from the infected macrophage and replication in the extracellular milieu. Activation of effector caspases was evident as green fluorescence at ~ 11h *p.i.* Shortly thereafter, caspase activation was detectable in uninfected bystander cells. In contrast, JE2 *brnQ1* did not replicate, remained confined inside macrophages and failed to trigger caspase activation during the assayed time (i.e., up to 20h *p.i.*) (S1 Movie).

In non-professional phagocytes, such as epithelial cells, JE2 *brnQ1* exhibited similarly impaired intracellular replication, accompanied by reduced cytotoxicity against the host, despite major differences in infection dynamics in this cell type [i.e., bacterial phagosomal escape precedes cytosolic replication [7,37]] (S4 and S5 Figs and S2–S5 Movies and S1 Text).

Notably, culture supernatants from *brnQ1* bacteria are as cytotoxic as JE2 WT supernatants, indicating that in principle, BrnQ1 is dispensable for secreted toxin production if bacteria grow in a rich medium (S6a Fig). Additionally, *brnQ1* mutants show wild-type haemolysis, on sheep blood agar, which supports the observation that *brnQ1* mutants can be virulent under the right growth conditions (S6b and S6c Fig).

Next, we analysed infected macrophages by transmission electron microscopy (TEM). We observed that, at 10h *p.i.*, both JE2 WT and JE2 *brnQ1* were localised to the lumen of intact phagosomal compartments (Fig 1g). Overall, these data demonstrate that wild-type *S. aureus* overcomes intra-macrophage confinement whereas the *brnQ1* mutant cannot. Moreover, under our experimental conditions, 10h *p.i.* is a timepoint at which both wild-type and the *brnQ1* mutant, which fails to replicate, are still confined within membrane-bound phagosomes.

## Transcriptome analysis of intracellular *S. aureus* reveals dysregulation of iron metabolism in *brnQ1* mutants

To identify the cause of the restricted intracellular growth of *brnQ1* mutants, we next performed dual RNA sequencing (RNAseq) of infected macrophages at 10h *p.i.* We hypothesized that this time point would be crucial for the bacterial adaptation to the phagolysosomal environment, since *S. aureus* JE2 WT and *brnQ1* mutant still reside within a membrane-bound vacuole but only the WT strain can initiate replication shortly thereafter. Consistent with our previous observations [38], we detected a similar host gene expression signature, irrespective of whether macrophages were infected with JE2 *brnQ1* or JE2 WT (Figs 2a and S7a), In contrast, we detected profound differences in gene expression between JE2 WT and JE2 *brnQ1*. Among bacterial transcripts, we found that 80 genes were significantly downregulated in the JE2 *brnQ1* mutant inside macrophages as compared to WT. Only 6 transcripts were upregulated in JE2 *brnQ1* (Figs 2b and S7b and Table 1). Most of the downregulated genes in intracellular JE2 *brnQ1*, belonged to operons that encode iron acquisition systems (Figs 2c and S8). These included several *isd* heme utilization genes (*isdA*, *isdB*, *isdCDEF* and *isdG*), *srtB*, staphyloferrin B siderophore biosynthesis genes (*sirAB* and *sbnABCDEFGH*) and the staphyloferrin A biosynthesis gene *sfaC* (Table 1). Several other genes, including chemotaxis-inhibiting protein (*chp*), thermonuclease (*nuc*), von Willebrand factor-binding protein (*vwb*), superoxide dismutase (*sodM*), Panton-Valentine leukocidin (PVL) subunit *lukS*, were downregulated. The cysteine protease staphopain A (*scpA*), as well as genes encoding superantigen-like proteins Ssl2, 4, 7, 11, 12, 13, 14, which are known virulence factors, were similarly downregulated as compared to wild-type.

We further validated by RT-qPCR the marked downregulation of heme- and siderophore-mediated iron acquisition genes in intracellular JE2 *brnQ1* as compared to intracellular WT JE2 (Fig 2d). It is interesting to note that a few genes showed modest upregulation in the *brnQ1* mutant (Table 1). Again, using RT-qPCR we confirmed that, in the *brnQ1* mutant, two loci in the *opp3* oligopeptide transporter operon [26] and a gene which encodes a putative transporter for *p*-aminobenzoyl-glutamate (PABA) (SAUSA300_RS13380/ SAUSA300_2417), a precursor of folic acid [28], were indeed upregulated in intracellular JE2 *brnQ1* (Fig 2e). Importantly, all changes in gene expression observed in JE2 *brnQ1* relative to JE2 WT were restored to wild-type levels when JE2 *brnQ1* was complemented with *brnQ1* on a plasmid (Fig 2d and 2e).

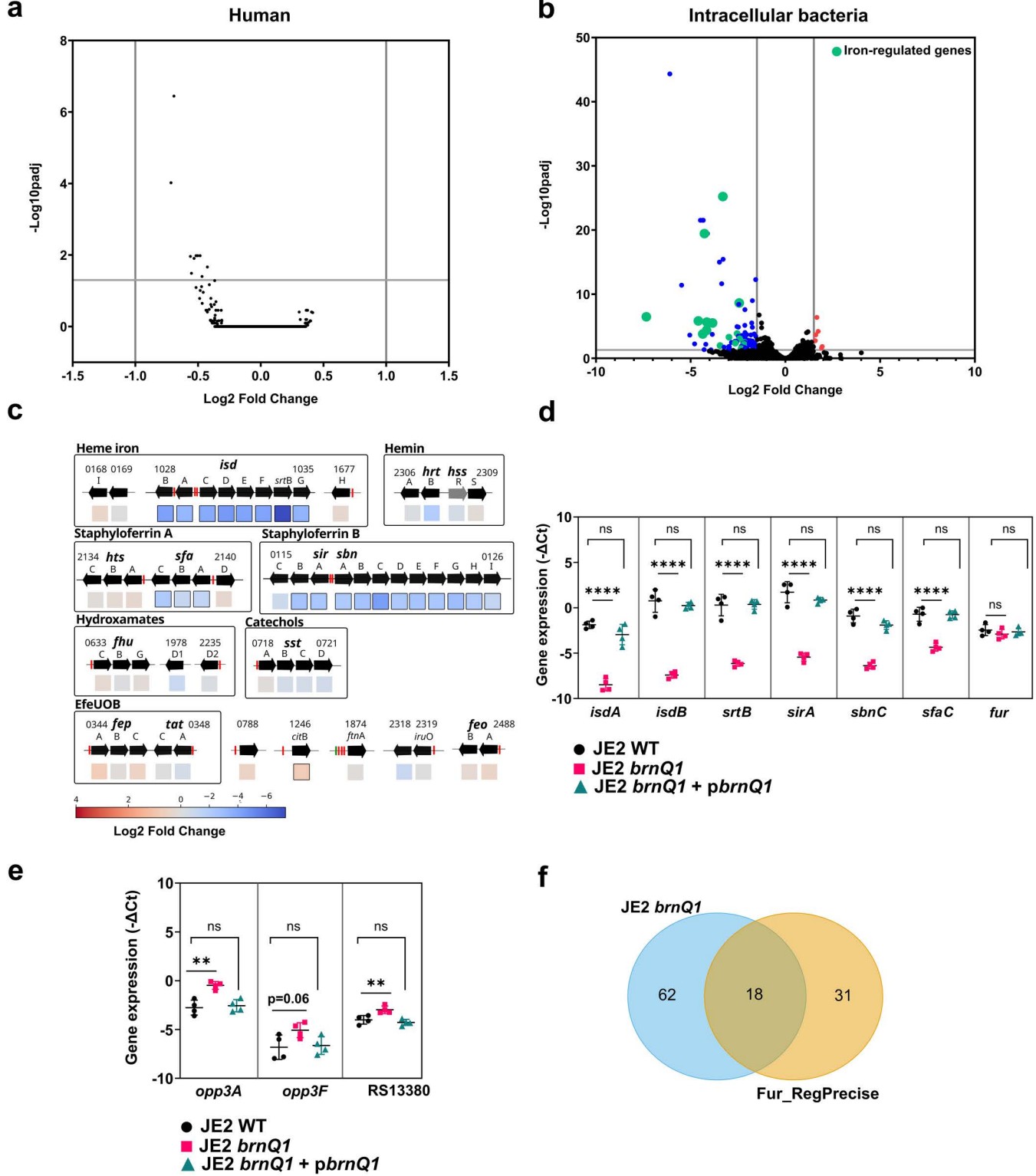

**Fig 2. Intracellular *S. aureus* JE2 *brnQ1* mutants do not respond to iron-starvation inside macrophages.** Dual RNAseq of primary human macrophages infected with either JE2 WT or JE2 *brnQ1*, at 10h *p.i.*: **(a)** Human transcripts and **(b)** Bacterial transcripts. Horizontal lines represent a p-value cutoff < 0.05. Vertical lines represent cutoffs of $\log_2$ fold changes of either <-1 and > 1 for human transcripts or <-1.5 and > 1.5 for bacterial transcripts.

Transcripts of iron uptake systems, which are prominently downregulated in intracellular JE2 *brnQ1* vs JE2 WT, are depicted in green. Differential transcript abundance is expressed as JE2 *brnQ1* vs JE2 WT (DESeq2). Data are shown from 3 independent experiments (n = 3). **(c)** Schematic of the ferric uptake regulator Fur regulon of *S. aureus* (information retrieved from RegPrecise). Coloration of the boxes underneath the open reading frames indicates $\log_2$ fold changes in expression of JE2 *brnQ1* vs JE2 WT according to the heatmap. Gene identifiers correspond to USA300_FPR3757 (old locus ID). **(d)** RT-qPCR analysis of selected genes found to be downregulated or upregulated **(e)** in intracellular JE2 *brnQ1*. Data are shown as mean -ΔCt values ±SD from independent experiments (n = 4, different donors). **(f)** Venn diagram showing the overlap between the set of significantly downregulated genes in JE2 *brnQ1* shown in **b** and the Fur regulon retrieved from RegPrecise. Statistical analysis: (**d**, **e**) one-way ANOVA with Dunnett's multiple comparisons test (vs JE2 WT for each gene); *ns = not significant; \*\*p < 0.01; \*\*\*\*p < 0.0001*.

Taken together, these data indicate that wild-type *S. aureus* can adapt to the intracellular milieu of the macrophage as evidenced, for instance, by upregulation of many genes including Fe-regulated loci, while the *brnQ1* mutant displays a global defect in gene expression.

The *S. aureus*-containing phagosome has been shown to be iron depleted [39]. By comparing transcriptomes of intracellular bacteria at 10h *p.i.* to bacteria grown for 10h in *Infection Medium,* we reaffirm the observation that *S. aureus* is indeed iron-starved in this niche, as indicated by upregulation of iron-regulated loci by intracellular JE2 WT (S9 Fig).

In *S. aureus,* transcription of genes encoding iron uptake proteins is controlled by the ferric uptake regulator Fur [40]. Fur is a transcription factor that uses $Fe^{2+}$ as corepressor. Under conditions of iron starvation, when Fur is not bound to $Fe^{2+}$, Fur is inactive resulting in de-repression of target genes [41]. The observation that many of the most significantly downregulated genes in the *brnQ1* mutant function in iron acquisition, prompted us to consult a list of Fur-regulated targets retrieved from the RegPrecise database [42]. This analysis revealed that 18 of the 80 (22.5%) genes that were significantly downregulated in JE2 *brnQ1* indeed belonged to the Fur regulon (Fig 2f). To exclude that inadvertent mutations in *fur* were responsible for our observations, we sequenced the *fur* locus from JE2 WT and JE2 *brnQ1* and found that the gene was intact in both strains.

### Intracellular *S. aureus brnQ1* mutants are metabolically quiescent at 10h *p.i.* and fail to respond to environmental cues

Since the *brnQ1* mutant carries an intact *fur* gene, yet fails to regulate iron acquisition loci, we speculated that the *brnQ1* mutant may lack the ability to respond appropriately to environmental cues within the phagosome. Ostensibly this defect is exclusively due to BCAA starvation. Therefore, we posited that, even upon inactivation of the *fur* gene, Fur-regulated genes would still not be de-repressed in the intracellular environment.

To test this, we generated *fur* and *brnQ1/fur* mutants in the JE2 background. To validate our strains, we first analysed *isdB* expression, as a representative iron-regulated gene within the Fur regulon, in *Infection Medium* (iron replete; contains BCAAs). Under this condition, the inactivation of *fur* resulted in increased expression of *isdB*, that was comparable between both *fur* mutant backgrounds (Fig 3a). Moreover, when we cultivated both JE2 *fur* and JE2 *brn*Q1/*fur* in iron-rich medium (RPMI1640 supplemented with 20 μM $FeSO_4$), total intracellular iron ($Fe^{2+}$ and $Fe^{3+}$) was ~ 3-fold higher for both *fur* backgrounds as compared to either JE2 WT or JE2 *brnQ1* (S10 Fig). This confirms that inactivation of *fur*, in both wild-type and mutant backgrounds, results in constitutive de-repression of iron-regulated operons in planktonic bacteria.

Using these strains, we analysed transcription of *isdB* inside macrophages and observed that JE2 WT and JE2 *fur* show comparable *isdB* expression. This result was expected as under the iron-starved conditions encountered within the phagosome, *isdB* should be fully de-repressed in JE2 WT similarly to JE2 *fur*, where iron-regulated genes are constitutively expressed. Intracellular JE2 *brnQ1/fur* showed *isdB* expression levels similar to JE2 *brnQ1* alone despite lacking the Fur repressor (Fig 3b). Furthermore, *isdB* expression in both *brnQ1* and *brnQ1/fur* mutants was significantly decreased relative to JE2 WT or JE2 *fur,* again indicating that *brnQ1* deficiency is enough to prevent appropriate induction of gene expression (Fig 3b).

The inability of JE2 *brnQ1* to remodel its gene expression in the intracellular milieu hinted that the bacteria may enter a state of intracellular *quiescence.* To test this, we assessed whether JE2 *brnQ1* bacteria were capable of *de novo* protein synthesis within macrophages. We measured *de novo* synthesis of the cyan-fluorescent protein *Cerulean* upon

**Table 1. List of regulated genes for *S. aureus* JE2 in primary human macrophages at 10h *p.i.* (JE2 *brnQ1* compared to JE2 WT).**

**Downregulated Genes**

| Locus ID (old) | Locus ID (new) | Gene | Annotation | Log$_2$FC | Adj. P value |
|---|---|---|---|---|---|
| SAUSA300_1034 | SAUSA300_RS05565 | srtB | sortase B | -7.34 | $3.38 \times 10^{-07}$ |
| SAUSA300_1920 | SAUSA300_RS10530 | chp | chemotaxis-inhibiting protein CHIPS | -6.10 | $4.67 \times 10^{-45}$ |
| SAUSA300_0407 | SAUSA300_RS02185 | ssl11 | superantigen-like protein SSL11 | -5.48 | $4.10 \times 10^{-12}$ |
| SAUSA300_0398 | SAUSA300_RS02130 | ssl4 | superantigen-like protein SSL4 | -5.05 | $2.38 \times 10^{-04}$ |
| SAUSA300_2291 | SAUSA300_RS12660 | gltS | sodium/glutamate symporter | -4.80 | $5.66 \times 10^{-03}$ |
| SAUSA300_1031 | SAUSA300_RS05550 | isdD | hypothetical protein | -4.59 | $1.49 \times 10^{-06}$ |
| SAUSA300_0408 | SAUSA300_RS02190 | – | FKLRK protein | -4.48 | $2.99 \times 10^{-22}$ |
| SAUSA300_0401 | SAUSA300_RS02150 | ssl7 | superantigen-like protein SSL7 | -4.37 | $2.99 \times 10^{-22}$ |
| SAUSA300_0120 | SAUSA300_RS00620 | sbnC | staphyloferrin B biosynthesis protein SbnC | -4.36 | $1.61 \times 10^{-04}$ |
| SAUSA300_0774 | SAUSA300_RS04175 | emp | extracellular matrix protein-binding adhesin Emp | -4.33 | $2.99 \times 10^{-22}$ |
| SAUSA300_1022 | SAUSA300_RS05500 | – | hypothetical protein | -4.29 | $4.43 \times 10^{-02}$ |
| SAUSA300_1028 | SAUSA300_RS05535 | isdB | heme uptake protein IsdB | -4.28 | $3.56 \times 10^{-20}$ |
| SAUSA300_0940 | SAUSA300_RS05050 | – | DoxX family protein | -4.20 | $6.60 \times 10^{-03}$ |
| SAUSA300_1032 | SAUSA300_RS05555 | isdE | heme ABC transporter substrate-binding protein IsdE | -4.16 | $4.45 \times 10^{-05}$ |
| SAUSA300_1033 | SAUSA300_RS05560 | isdF | iron ABC transporter permease | -4.15 | $2.27 \times 10^{-06}$ |
| SAUSA300_0775 | SAUSA300_RS04180 | – | hypothetical protein | -4.12 | $3.56 \times 10^{-20}$ |
| SAUSA300_1929 | SAUSA300_RS10585 | – | phage tail protein | -3.93 | $4.86 \times 10^{-02}$ |
| SAUSA300_0776 | SAUSA300_RS04185 | nuc | thermonuclease family protein | -3.86 | $1.85 \times 10^{-04}$ |
| SAUSA300_1030 | SAUSA300_RS05545 | isdC | heme uptake protein IsdC | -3.85 | $3.09 \times 10^{-06}$ |
| SAUSA300_0815 | SAUSA300_RS04395 | ear | DUF4888 domain-containing protein | -3.48 | $1.03 \times 10^{-15}$ |
| SAUSA300_1035 | SAUSA300_RS05570 | isdG | staphylobilin-forming heme oxygenase IsdG | -3.45 | $1.04 \times 10^{-02}$ |
| SAUSA300_0215 | SAUSA300_RS01130 | – | isoprenylcysteine carboxyl methyltransferase family protein | -3.36 | $2.36 \times 10^{-12}$ |
| SAUSA300_1029 | SAUSA300_RS05540 | isdA | LPXTG-anchored heme-scavenging protein IsdA | -3.30 | $6.07 \times 10^{-26}$ |
| SAUSA300_0773 | SAUSA300_RS04170 | vwb | von Willebrand factor binding protein Vwb | -3.29 | $3.62 \times 10^{-16}$ |
| SAUSA300_1060 | SAUSA300_RS05745 | ssl13 | superantigen-like protein SSL13 | -3.23 | $1.84 \times 10^{-02}$ |
| – | SAUSA300_RS09430 | – | hypothetical protein | -3.01 | $4.62 \times 10^{-02}$ |
| SAUSA300_2528 | SAUSA300_RS14025 | – | epoxyqueuosine reductase QueH | -2.99 | $1.44 \times 10^{-02}$ |
| SAUSA300_0124 | SAUSA300_RS00640 | sbnG | staphyloferrin B biosynthesis citrate synthase SbnG | -2.98 | $5.10 \times 10^{-04}$ |
| SAUSA300_1061 | SAUSA300_RS05750 | ssl14 | superantigen-like protein SSL14 | -2.97 | $1.44 \times 10^{-02}$ |
| SAUSA300_1059 | SAUSA300_RS05740 | ssl12 | superantigen-like protein SSL12 | -2.96 | $1.82 \times 10^{-02}$ |
| SAUSA300_0237 | SAUSA300_RS01265 | – | nucleoside hydrolase | -2.93 | $4.86 \times 10^{-02}$ |
| SAUSA300_0119 | SAUSA300_RS00615 | sbnB | 2,3-diaminopropionate biosynthesis protein SbnB | -2.65 | $4.24 \times 10^{-03}$ |
| SAUSA300_0116 | SAUSA300_RS00600 | sirB | staphyloferrin B ABC transporter permease subunit SirB | -2.62 | $2.94 \times 10^{-03}$ |
| SAUSA300_0986 | SAUSA300_RS05305 | cydA | cytochrome ubiquinol oxidase subunit I | -2.55 | $1.14 \times 10^{-05}$ |
| SAUSA300_0122 | SAUSA300_RS00630 | sbnE | lucA/IucC family siderophore biosynthesis protein | -2.55 | $1.35 \times 10^{-04}$ |
| SAUSA300_2453 | SAUSA300_RS13605 | – | ATP-binding cassette domain-containing protein | -2.49 | $3.75 \times 10^{-04}$ |
| SAUSA300_1786 | SAUSA300_RS09780 | ecsA | ABC transporter ATP-binding protein | -2.48 | $1.37 \times 10^{-05}$ |
| SAUSA300_0814 | SAUSA300_RS04390 | – | Abi family protein | -2.46 | $3.80 \times 10^{-09}$ |
| SAUSA300_0117 | SAUSA300_RS00605 | sirA | staphyloferrin B ABC transporter substrate-binding protein SirA | -2.44 | $2.38 \times 10^{-09}$ |
| SAUSA300_0123 | SAUSA300_RS00635 | sbnF | lucA/IucC family siderophore biosynthesis protein | -2.42 | $9.01 \times 10^{-04}$ |
| SAUSA300_0307 | SAUSA300_RS01635 | – | 5'-nucleotidase, lipoprotein e(P4) family | -2.41 | $4.79 \times 10^{-04}$ |
| SAUSA300_0902 | SAUSA300_RS04855 | pepF | oligoendopeptidase F | -2.38 | $2.11 \times 10^{-09}$ |

*(Continued)*

**Table 1.** (Continued)

**Downregulated Genes**

| Locus ID (old) | Locus ID (new) | Gene | Annotation | Log$_2$FC | Adj. P value |
|---|---|---|---|---|---|
| SAUSA300_0121 | SAUSA300_RS00625 | sbnD | staphyloferrin B export MFS transporter | -2.33 | $3.05 \times 10^{-02}$ |
| SAUSA300_2409 | SAUSA300_RS13340 | cnzC | nickel ABC transporter permease | -2.33 | $1.32 \times 10^{-02}$ |
| SAUSA300_1171 | SAUSA300_RS06335 | – | insulinase family protein | -2.30 | $2.78 \times 10^{-04}$ |
| SAUSA300_0825 | SAUSA300_RS04450 | – | nitronate monooxygenase | -2.26 | $1.19 \times 10^{-02}$ |
| SAUSA300_1675 | SAUSA300_RS09145 | tyrS | tyrosine-tRNA ligase | -2.25 | $4.74 \times 10^{-03}$ |
| SAUSA300_0118 | SAUSA300_RS00610 | sbnA | siderophore biosynthesis protein SbnA | -2.20 | $1.29 \times 10^{-02}$ |
| SAUSA300_2137 | SAUSA300_RS11770 | sfaC | staphyloferrin A biosynthesis protein SfaC | -2.19 | $6.69 \times 10^{-03}$ |
| SAUSA300_2262 | SAUSA300_RS12500 | spdB | CPBP family intramembrane metalloprotease SdpB | -2.17 | $1.92 \times 10^{-02}$ |
| SAUSA300_0125 | SAUSA300_RS00645 | sbnH | staphyloferrin B biosynthesis decarboxylase SbnH | -2.16 | $3.36 \times 10^{-03}$ |
| SAUSA300_1053 | SAUSA300_RS05680 | flr | formyl peptide receptor-like 1 inhibitory protein | -2.14 | $7.19 \times 10^{-06}$ |
| SAUSA300_0135 | SAUSA300_RS00705 | sodM | superoxide dismutase | -2.14 | $2.62 \times 10^{-08}$ |
| SAUSA300_1957 | SAUSA300_RS10735 | dnaD2 | DnaD domain-containing protein | -2.14 | $3.05 \times 10^{-02}$ |
| SAUSA300_1890 | SAUSA300_RS10340 | scpA | cysteine protease staphopain A | -2.13 | $5.77 \times 10^{-04}$ |
| SAUSA300_2251 | SAUSA300_RS12430 | – | octopine dehydrogenase | -2.13 | $9.70 \times 10^{-05}$ |
| SAUSA300_1852 | SAUSA300_RS10120 | – | ABC transporter ATP-binding protein | -2.13 | $3.36 \times 10^{-03}$ |
| SAUSA300_1964 | SAUSA300_RS10775 | – | DUF771 domain-containing protein | -2.10 | $2.45 \times 10^{-02}$ |
| SAUSA300_1587 | SAUSA300_RS08650 | hisS | histidine-tRNA ligase | -2.04 | $2.03 \times 10^{-04}$ |
| SAUSA300_1797 | SAUSA300_RS09835 | – | helix-turn-helix transcriptional regulator | -1.99 | $5.26 \times 10^{-03}$ |
| SAUSA300_0396 | SAUSA300_RS02115 | ssl2 | superantigen-like protein SSL2 | -1.99 | $1.19 \times 10^{-02}$ |
| SAUSA300_0254 | SAUSA300_RS01355 | lytS | sensor histidine kinase | -1.98 | $1.46 \times 10^{-03}$ |
| SAUSA300_0987 | SAUSA300_RS05310 | cydB | cytochrome D ubiquinol oxidase subunit II | -1.85 | $1.67 \times 10^{-03}$ |
| SAUSA300_2537 | SAUSA300_RS14075 | ldh2 | L-lactate dehydrogenase | -1.82 | $2.29 \times 10^{-04}$ |
| SAUSA300_0432 | SAUSA300_RS02315 | – | sodium-dependent transporter | -1.81 | $6.37 \times 10^{-03}$ |
| SAUSA300_2210 | SAUSA300_RS12185 | glcU | glucose uptake protein GlcU | -1.79 | $1.06 \times 10^{-02}$ |
| SAUSA300_1382 | SAUSA300_RS07545 | lukS | Panton-Valentine bi-component leukocidin subunit LukS-PV | -1.78 | $3.12 \times 10^{-06}$ |
| SAUSA300_1882 | SAUSA300_RS10300 | gatC | Asp-tRNA(Asn)/Glu-tRNA(Gln) amidotransferase subunit GatC | -1.75 | $2.00 \times 10^{-03}$ |
| SAUSA300_1381 | SAUSA300_RS07540 | – | Panton-Valentine bi-component leukocidin subunit F | -1.75 | $1.48 \times 10^{-05}$ |
| SAUSA300_1471 | SAUSA300_RS08030 | xseB | exodeoxyribonuclease VII small subunit | -1.74 | $3.90 \times 10^{-02}$ |
| – | SAUSA300_RS10500 | – | MAP domain-containing protein | -1.74 | $1.04 \times 10^{-09}$ |
| SAUSA300_2526 | SAUSA300_RS14015 | pyrD | quinone-dependent dihydroorotate dehydrogenase | -1.73 | $2.21 \times 10^{-03}$ |
| SAUSA300_2250 | SAUSA300_RS12425 | nhaC | Na + /H+ antiporter NhaC | -1.69 | $3.16 \times 10^{-02}$ |
| SAUSA300_2328 | SAUSA300_RS12865 | – | DUF4889 domain-containing protein | -1.67 | $2.90 \times 10^{-02}$ |
| SAUSA300_0107 | SAUSA300_RS00555 | nptA | Na/Pi cotransporter family protein | -1.65 | $1.61 \times 10^{-04}$ |
| SAUSA300_1153 | SAUSA300_RS06245 | uppS | isoprenyl transferase | -1.64 | $4.06 \times 10^{-03}$ |
| SAUSA300_1170 | SAUSA300_RS06330 | – | GntR family transcriptional regulator | -1.63 | $2.40 \times 10^{-02}$ |
| SAUSA300_0025 | SAUSA300_RS00130 | adsA | LPXTG-anchored adenosine synthase AdsA | -1.62 | $5.72 \times 10^{-03}$ |
| SAUSA300_0620 | SAUSA300_RS03325 | mntA | metal ABC transporter ATP-binding protein | -1.57 | $5.35 \times 10^{-13}$ |
| SAUSA300_0669 | SAUSA300_RS03590 | uppP | undecaprenyl-diphosphate phosphatase | -1.56 | $3.92 \times 10^{-04}$ |
| ***Upregulated Genes*** | | | | | |
| Locus ID (old) | Locus ID (new) | Gene | Annotation | Log$_2$FC | Adj. P value |
| SAUSA300_2417 | SAUSA300_RS13380 | – | AbgT family transporter | 1.58 | $1.83 \times 10^{-03}$ |
| SAUSA300_0955 | SAUSA300_RS05135 | atl | bifunctional autolysin | 1.60 | $2.24 \times 10^{-04}$ |

*(Continued)*

**Table 1.** (Continued)

**Downregulated Genes**

| Locus ID (old) | Locus ID (new) | Gene | Annotation | Log$_2$FC | Adj. P value |
|---|---|---|---|---|---|
| SAUSA300_0891 | SAUSA300_RS04805 | *opp-3A* | peptide ABC transporter substrate-binding protein | 1.66 | 4.22 x 10$^{-07}$ |
| SAUSA300_0890 | SAUSA300_RS04800 | *opp-3F* | ATP-binding cassette domain-containing protein | 1.73 | 6.41 x 10$^{-05}$ |
| SAUSA300_2585 | SAUSA300_RS14375 | *asp3* | accessory Sec system protein Asp3 | 1.91 | 2.45 x 10$^{-02}$ |
| SAUSA300_2607 | SAUSA300_RS14495 | *hisA* | 1-(5-phosphoribosyl)-5-((5-phosphoribosylamino) methylideneamino)imidazole-4-carboxamide isomerase | 1.94 | 1.44 x 10$^{-02}$ |

intracellular induction with anhydrotetracycline (aTc) as previously described [43]. In a control experiment, both JE2 WT and JE2 *brnQ1* responded to aTc induction in TSB medium, showing that, in principle, both strains can produce *Cerulean* (S11 Fig). To test whether *Cerulean* production can be induced by intracellular bacteria, aTc was added to infected macrophages at 1h or 10h *p.i.* for 2h (Fig 3c). Induction with aTc at 1h *p.i.* resulted in *Cerulean* fluorescence for both JE2 WT and JE2 *brnQ1* (Fig 3d). At this time, approximately 15% of all internalized bacteria, as detected by BODIPY FL Vancomycin staining, also showed *Cerulean* fluorescence, while in vehicle-treated cells the bacteria remained *Cerulean* negative (Figs 3e and S12). In contrast, when *Cerulean* production was induced at 10h *p.i.*, ~10% of the JE2 WT bacteria became *Cerulean*-positive, whereas only very rarely did JE2 *brnQ1* become *Cerulean*-positive and was comparable to vehicle treated controls (Figs 3f, 3g and S12).

The inactivation of *brnQ1* alone, the dominant BCAA transporter, was sufficient to halt intracellular bacterial growth. This indicated that the *S. aureus*-containing phagosome must be scarce in BCAAs. To test this directly, we supplemented the *Infection Medium* with 1mM of each BCAA and observed that the defect in intracellular replication was rescued, both in the case of JE2 *brnQ1* (Fig 3h) and JE2 *brnQ1*/*fur* (Fig 3i), which now overcame macrophage restriction and grew unrestrictedly. Moreover, addition of excess BCAAs rendered intracellular JE2 *brnQ1* and JE2 *brnQ1*/*fur* responsive to the iron-deplete state of the phagosome, as shown by significantly higher *isdB* expression levels 10h *p.i.* (Fig 3j).

During growth in CDM lacking all three BCAAs, the growth of *S. aureus* is preceded by an extended lag phase of ~20h. However, omission of Val and Leu only, surprisingly completely halted bacterial growth (S13a Fig). We have therefore hypothesized that provision of these 2 amino acids alone would be sufficient to rescue the growth deficit of *S. aureus brnQ1* in macrophages. Indeed, supplementation with 1mM Val and Leu, but not the single amino acids, restored growth of the *brnQ1* mutant in RAW 264.7 macrophages (S13b and S13c Fig).

Taken together, we posit that by 10h *p.i. S. aureus brnQ1* mutants have assumed a state of *quiescenc*e where the bacteria fail to adapt to the intracellular niche as evidenced by altered gene expression and impaired *de novo* protein synthesis. Importantly, this *quiescence* is caused by BCAA starvation and can be overcome if Val and Leu become available, indicating that BCAA acquisition underpins *S. aureus* growth and survival inside macrophages.

## Inactivation of the CodY repressor rescues growth of *brnQ1*-deficient *S. aureus* in an IlvD and BcaP dependent manner

The *brnQ1* mutant fails to grow in BCAA-depleted phagosomes, although *S. aureus* carries genes that encode BCAA biosynthesis enzymes (e.g., the *ilv* operon) [21], as well as a second BCAA transporter (*bcaP*) [29], which are expressed upon inactivation of the repressor protein CodY [21, 26, 29, 44] and could potentially ensure BCAA supply.

To understand the role that CodY plays in the acquisition of BCAAs during infection, we created a *brnQ1*/*codY* double mutant and measured bacterial replication in macrophages. In contrast to the *brnQ1* single mutant, the *brnQ1*/*codY* double mutant behaved similarly to the WT strain and exhibited *URG* both in primary human (Fig 4a) and murine RAW 264.7 macrophages (Fig 4b). Restored intracellular growth of *brnQ1*-deficient *S. aureus* upon *codY* inactivation, indicated that the bacteria must now be able to meet their BCAA requirement.

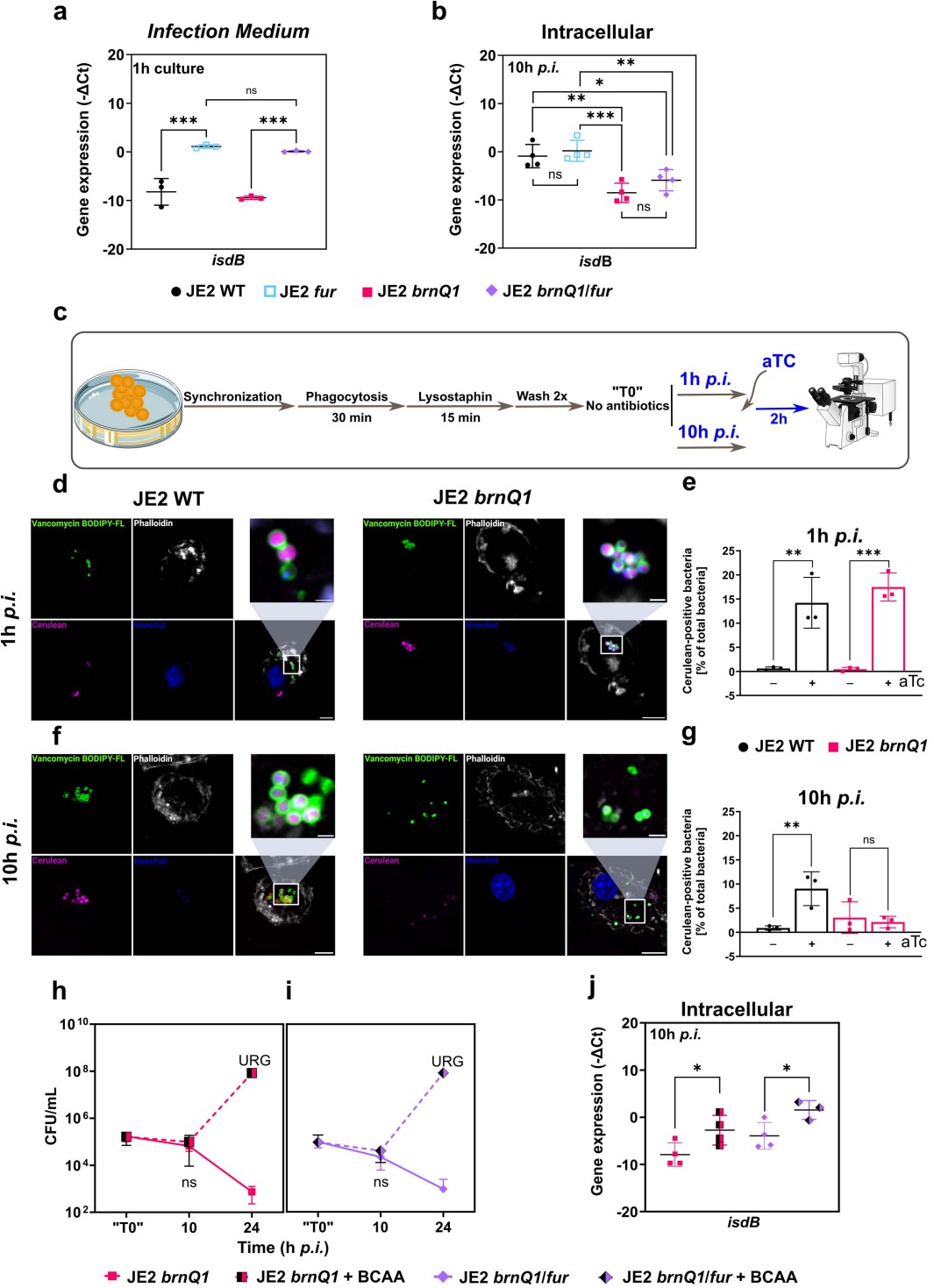

**Fig 3. Loss of BrnQ1 function renders intracellular *S. aureus* unresponsive to environmental cues and metabolically quiescent, but a supply of exogenous BCAAs restores responsiveness and promotes intracellular replication.** **(a)** RT-qPCR analysis of *isdB* expression for bacteria grown in *Infection Medium* for 1 hour or **(b)** in intracellular bacteria within infected primary human macrophages, at 10h *p.i. isdB* was selected as representative iron-regulated gene to report on Fur regulon status. Data are shown as mean -ΔCt values ±SD from independent experiments: (a) n = 3; (b) n = 4. **(c)** Schematic representation of the set-up for experiments using aTc induction of intracellular JE2 with a plasmid allowing anhydrous tetracycline

(aTc)-inducible expression of the cyan-fluorescent protein *Cerulean*. After 1h *p.i.* or 10h *p.i.*, infected cells were treated with either vehicle control (-) or 300 ng/mL aTc (+) for 2h to allow *Cerulean* expression. **(d)** Representative confocal images of infected primary human macrophages at 1h *p.i.* and **(f)** 10h *p.i.* upon induction (Magenta: *Cerulean*; Green: Vancomycin BODIPY-FL/*S. aureus*; Gray: Phalloidin/Actin; Blue: Hoechst/DNA. Scale bar: overviews: 5 µm; zoomed inset: 1µm). **(e)** Quantification of intracellular *Cerulean*-positive bacteria, as percent of total bacteria at 1h *p.i.* and **(g)** 10h *p.i.* in samples exposed to aTc (+) or vehicle control (-) for 2h. Data are shown as mean ±SD from independent experiments (n = 3). **(h)** Intracellular replication of JE2 *brnQ1* or **(i)** JE2 *brnQ1/fur* in primary human macrophages, without or with BCAA supplementation. Excess BCAAs (1mM each) were supplemented 24h prior to and during the infection. Data are shown as mean ±SD from independent experiments (n = 3). Graphed *URG* value represents an imputed value (median of all quantified *URG* experiments which are shown in Fig 1d). **(j)** RT-qPCR analysis of *isdB* expression in intracellular bacteria in infected primary human macrophages, at 10h *p.i.* Excess BCAAs (1mM each) were supplemented 24h prior to and during the infection. Data are shown as mean -ΔCt values ±SD from independent experiments: n = 4; except JE2 *brnQ1/fur*+BCAA: n = 3. Statistical analysis: **(a, b)** one-way ANOVA with Tukey's multiple comparisons test; **(e, g, h, i, j)** one-way ANOVA with Sidak's multiple comparisons test (h and i: for the 10h *p.i.* time point only); *ns = not significant; *p < 0.05; **p < 0.01; ***p < 0.001; (URG = unrestricted growth)*. **(c)** contains icons which were modified (colour and design) from https://bioicons.com/, as follows: petri-dish-lid-yellow icon by Servier https://smart.servier.com/ is licensed under CC-BY 3.0 Unported https://creativecommons.org/licenses/by/3.0/. The confocal-scanning-laser-microscope-CSLM icon by DBCLS https://togotv.dbcls.jp/en/pics.html is licensed under CC-BY 4.0 Unported https://creativecommons.org/licenses/by/4.0/.

Conceivably, this could be accomplished through endogenous BCAA synthesis. To test this, we created a *brnQ1/codY/ilvD* triple mutant and assessed its ability to grow in CDM without Val. Growth experiments revealed that the *brnQ1/codY* mutant, as expected, could grow in the absence of exogenous Val yet the *brnQ1/codY/ilvD* mutant could not (S14 Fig). This confirms that, as opposed to the triple mutant, the *brnQ1/codY* mutant must be able to synthesize Val to allow growth in Val-deplete media.

Next, we infected RAW 264.7 macrophages with the mutants. As expected, *brnQ1*-deficient *S. aureus* failed to grow out of macrophages whereas WT and *brnQ1/codY S. aureus* overcame macrophage restriction and displayed *URG*. Remarkably, the *brnQ1/codY/ilvD* mutant also grew out of macrophages indicating that, despite the inability to synthesize BCAAs, the bacteria still access these amino acids inside the macrophage (Fig 4c).

The observation that the intracellular growth of *S. aureus brnQ1* can be restored by exogenous addition of Val and Leu during infection, indicated that the bacteria must retain the ability for BCAA uptake. BcaP is an obvious candidate as it has previously been shown that it transports all three BCAAs and that it can partially compensate for loss of *brnQ1* [29]. Therefore, we next infected macrophages with a *brnQ1* mutant strain that overexpresses *bcaP* or a respective vector control and we observed that *S. aureus brnQ1* overexpressing *bcaP* could grow unrestrictedly without BCAA supplementation (Fig 4d). These data suggested that increased *bcaP* expression, due to *codY* inactivation [29], could explain why the *brnQ1/codY/ilvD* triple mutant maintained the ability to overcome macrophage restriction. Hence, we utilized a *brnQ1/bcaP* double mutant, again inactivated either *codY* alone, or *codY* and *ilvD* in this background and infected RAW 264.7 macrophages. Similar to a *brnQ1* single mutant, the *brnQ1/bcaP* double mutant was unable to replicate (Fig 4e). Inactivation of *codY* in the *brnQ1/bcaP* background restored growth in macrophages however, an additional *ilvD* mutation significantly impaired growth (Fig 4e). Taken together, these data indicate that limited phagosomal BCAA levels can in principle be overcome in *S. aureus* by increased expression of CodY-regulated BCAA synthesis genes and/or BCAA transporters.

### *S. aureus brnQ1* mutants survive for inordinate duration within human macrophages

Since *brnQ1*-deficient *S. aureus* failed to replicate in macrophages during the first 24h of infection, we next assessed long-term intracellular survival in primary human macrophages. Often, studies on long-term intracellular *S. aureus* infections rely on sustained antibiotic treatment of the infected host cells to limit the effects caused by extracellular bacteria that may egress from host cells during the experiment [13,45–49]. Here we did not maintain antibiotic treatment on infected macrophages, and this allowed us to assess whether *URG* would occur for *S. aureus brnQ1.* Bacteria were quantified by CFU counting at 1, 7, 14, 21 and 28 days *p.i.* unless we observed *URG* (Fig 5a). Within the first 24h of infection, the number of JE2 *brnQ1* bacteria decreased to ~3.5% of the initially phagocytosed bacterial population (i.e., lysostaphin-protected bacteria at "T0"). After this initial decline, the abundance of JE2 *brnQ1* remained steady whereas JE2 WT grew

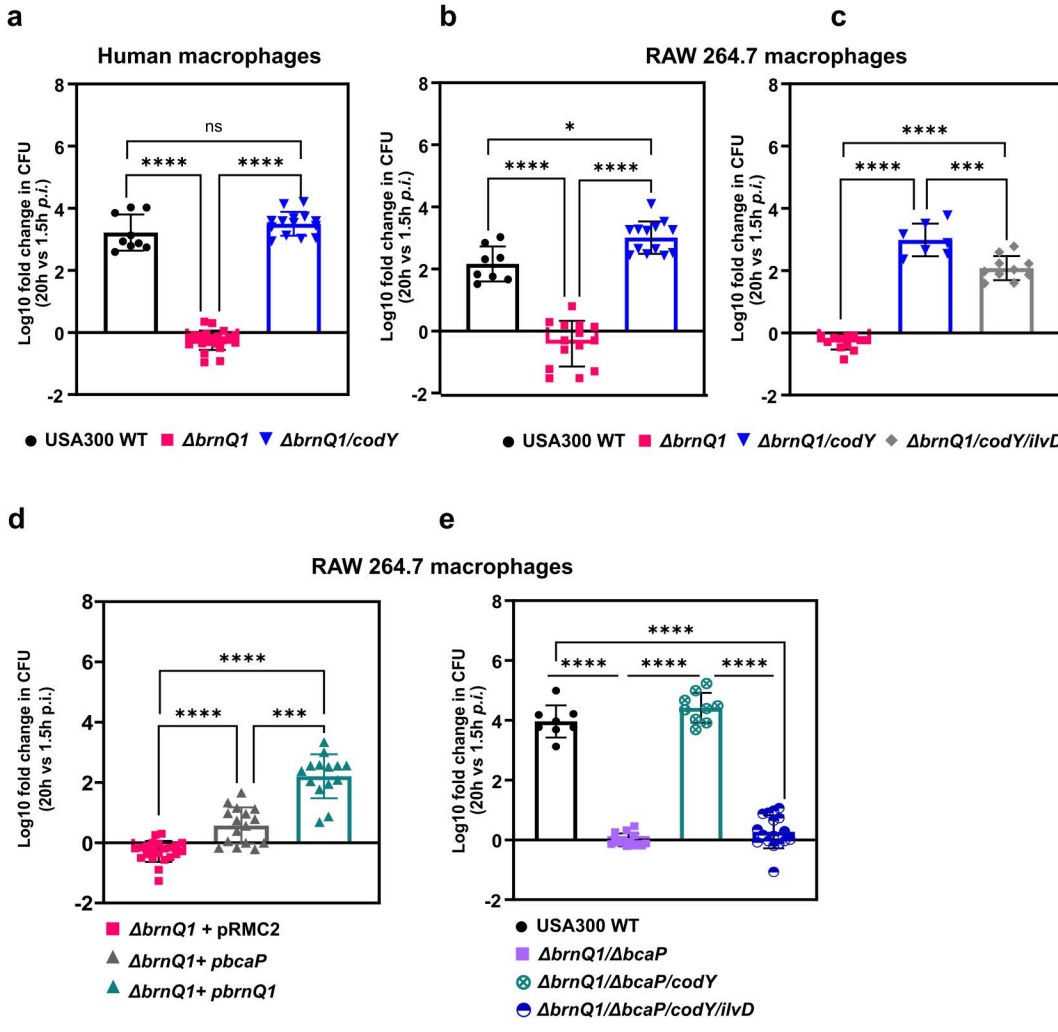

**Fig 4. Intracellular growth deficiency of *S. aureus brnQ1* mutants can be rescued by inactivation of the CodY repressor or BcaP overexpression.** Intracellular growth of a *brnQ1* mutant in the *S. aureus* USA300 background (Δ*brnQ1*) and a *brnQ1/codY* double mutant (Δ*brnQ1/codY*::Tn) in **(a)** primary human macrophages and **(b)** murine RAW 264.7 macrophages. **(c)** Intracellular growth in RAW264.7 macrophages of a triple *S. aureus* USA300 *brnQ1/codY/ilvD* mutant (Δ*brnQ1/codY*::Tn/*ilvD*::Tn-KanR). **(d)** Growth in infected RAW 264.7 macrophages of S. *aureus brnQ1* carrying either a vector control (pRMC2), plasmid-encoded BCAA transporter BcaP (Δ*brnQ1*+p*bcaP*) or plasmid encoded *brnQ1* (Δ*brnQ1*+p*brnQ1*). **(e)** Growth in infected RAW 264.7 macrophages of *S. aureus brnQ1*, a double *brnQ1/bcaP* mutant (Δ*brnQ1/ ΔbcaP*), a triple *brnQ1, bcaP, codY* mutant (Δ*brnQ1/ΔbcaP/codY*::Tn) and a quadruple mutant carrying the additional *ilvD* mutation (Δ*brnQ1/ΔbcaP/codY*::Tn/*ilvD*::Tn-KanR). USA300 WT serves as control. All data are shown as mean log10 value±S.D. for the calculated fold change in CFU/mL at 20h relative to 1.5h *p.i.*, for each bacterial strain. Each data point plotted represents a biological replicate derived from at least three independent experiments (n ≥ 3). Statistical analysis: (**a, b, c, e**) one-way ANOVA with Tukey's multiple comparison test; (**d**) Brown-Forsythe and Welch ANOVA with Dunnet's T3 multiple comparisons test. *ns = not significant; *p < 0.05, ***p < 0.001, ****p < 0.0001.*

unrestrictedly (*URG*) by day 1 *p.i.* (Fig 5b). Remarkably, viable JE2 *brnQ1* were recovered from infected macrophages at the endpoint of the assay after 28 days *p.i.* Across 14 independent experiments, we observed that typically JE2 *brnQ1* survived for 28 days within the macrophages. However, on occasion, *URG* was observed at day 2 or 15 *p.i.* albeit with less frequency (Fig 5b). Consistent with the CFU recovery data, only few bacteria were detected microscopically inside macrophages infected with JE2 *brnQ1* from 24h *p.i.* onwards and ultrastructural analysis revealed *brnQ1* mutants to be localised inside intact membrane-enclosed vacuoles (Fig 5c). We hypothesized that the variability in occurrence of *URG*

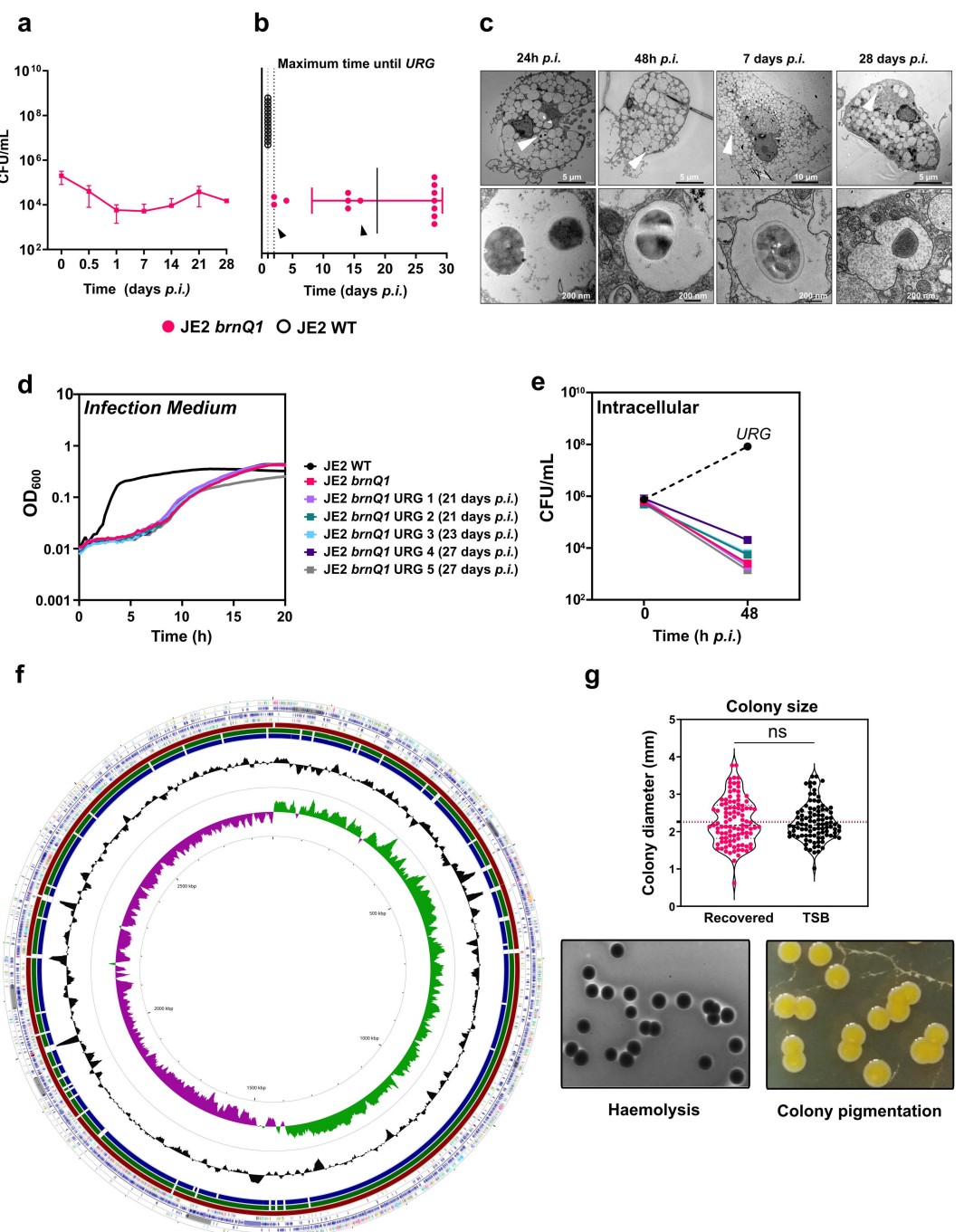

**Fig 5. *S. aureus brnQ1* mutants survive for at least 28 days inside macrophages but are distinct of *small colony variants* (SCV). (a)** JE2 *brnQ1* intracellular replication in primary human macrophages, assayed over a period of 28 days. Data are shown as mean values ±SD from independent experiments ($T_0$: n = 7; $T_{0.5}$: n = 5; $T_{d1}$: n = 6; $T_{d7}$ and $T_{d21}$: n = 2; $T_{d14}$ and $T_{d28}$: n = 3). **(b)** Maximum time until unrestricted growth (*URG*) in macrophages infected with JE2 *brnQ1* across 14 biological replicates (n = 14), using 13 different blood donors (arrows: same donor, but used in an independent experiment). JE2 WT serves as control (*URG* at 24h *p.i.*). The experiment was terminated 28 days *p.i.* regardless of *URG* occurrence. **(c)** Transmission electron microscopy of JE2 *brnQ1*-infected macrophages at 24h *p.i.*, 48h *p.i.*, 7 and 28 days *p.i.* (white arrows). Shown are representative micrographs from one experiment. **(d)** *URG* recovery experiment, whereby during a 28-day infection in macrophages, JE2 *brnQ1* clones (i.e., micro-colonies formed upon overgrowth from infected macrophages) were recovered upon *URG* (days 21, 23 and 27 *p.i.* respectively). Recovered JE2 *brnQ1* clones

were re-inoculated in *Infection Medium* and growth was analysed. OD$_{600}$ was measured every 18 min, for 20h. **(e)** Recovered JE2 *brnQ1* were used for infection experiments in primary human macrophages, for 48h. Graphed *URG* value for JE2 WT control, represents an imputed value (median of all quantified *URG* experiments which are shown in Fig 1d). **(f)** Circular genome alignment of two JE2 *brnQ1* clones recovered from macrophages upon *URG* (*URG 4* and *URG5* shown in **d** and **e** using the USA300_FPR3757 complete genome and JE2 WT for comparison (parental strain, laboratory stock). **(g)** JE2 *brnQ1* mutants that did not show *URG* were recovered from viable macrophages at 28 days *p.i.* and colony size, haemolysis on sheep-blood agar, and colony pigmentation were assessed. Colony diameter measurements of JE2 *brnQ1* which were recovered from macrophages, were compared with bacteria grown to late exponential phase in TSB. Data points correspond to single colonies, recovered from independent experiments (n = 2). Statistical analysis: **(g)** two-tailed Mann-Whitney test; *ns* = not significant.

might arise in a stochastic manner, resulting, for example, from loss of macrophage bactericidal capacity, death of individual host cells with subsequent release of bacteria, or heterogeneity within the bacterial subpopulations. To address this, we set up experiments where macrophages derived from the same donor were seeded in multiple technical replicates (e.g., up to 80 wells/experiment), infected with JE2 *brnQ1* and observed daily for 28 days to monitor *URG*. We observed variability within technical replicates, with *URG* occurring at various time points during the infection (S15a Fig). Furthermore, *brnQ1* mutants in different *S. aureus* backgrounds similarly survived for up to 28 days without eliciting visible cytotoxic effects against host cells (S15b, S15c and S15d Fig). These data collectively indicate that the duration of infection prior to *URG* is stochastic and donor-independent.

*S. aureus* can grow in the absence of Leu or Val *in vitro*, albeit after an extended lag phase (S13a Fig). Moreover, growth in the absence of Val was shown to select for heritable loss-of-function mutations in *codY* or the 5' untranslated region (5'UTR) of *ilvD,* to allow *de novo* BCAA synthesis and subsequent bacterial replication [21]. Spontaneous selection of such suppressor mutants in intracellular JE2 *brnQ1* is a possible scenario that could explain the differences in *URG* observed across experiments and/or technical replicates. Furthermore, our data have revealed that inactivation of CodY is sufficient to rescue the growth of *brnQ1 S. aureus* in macrophages and in CDM lacking Val (Figs 4 and S14a).

To determine whether *brnQ1* mutants that initiated *URG* had acquired genetic changes, we recovered five JE2 *brnQ1* clones upon egress from infected macrophages, sub-cultured the clones in *Infection Medium* and infected again macrophages. Each recovered *brnQ1* isolate again displayed delayed growth in *Infection Medium* (Fig 5d) and impaired growth upon macrophage infection (Fig 5e) suggesting that the egress events were not the result of heritable mutations. Consistently, whole genome sequencing of two such *brnQ1* isolates revealed no mutations in either *codY* or the 5'UTR of *ilvD* (Fig 5f).

## Intracellular *S. aureus brnQ1* mutants are not classical *persisters*

Within a bacterial subpopulation, transient growth arrest and metabolic quiescence, accompanied by antibiotic tolerance, are hallmarks of *persistence* [50–53]. Intracellular persisters of *S. aureus* can arise upon antibiotic pressure [43,54,55]. Several mechanisms triggering intracellular *S. aureus* persisters were proposed, such as a transient activation of the stringent response (SR) [43]. In *S. aureus*, SR can be induced by amino-acid starvation (especially Leu/Val deprivation [27]) and results in rapid synthesis of the (p)ppGpp alarmone leading to profound transcriptional reprogramming, especially the repression of proliferation-related genes and activation of the stress-response and starvation-related regulons and a halt of bacterial division [27]. However, upon consulting our transcriptomic data (S1 Dataset), we did not observe changes in the expression of translation factors, ribosomal proteins or tRNAs in JE2 *brnQ1* compared to wild-type, at 10h *p.i.* Similarly, the expression of *relA, relP* and *relQ,* encoding the bifunctional Rsh (p)ppGpp synthase/hydrolase (*relA*) and the RelP and RelQ synthases [56] remained unaltered (S8 Fig). A study found that persistence in intracellular *S. aureus* in the presence of antibiotics involved multiple protective mechanisms: transcriptional reprogramming of the cell wall stress stimulon, SOS response to preserve bacterial genome integrity or the heat-shock stimulon [43]. In case of intracellular JE2 *brnQ1*, however, we found none of these pathways to be differentially regulated (S1 Dataset and S8 Fig).

A particular type of persistence described for *S. aureus* is the so-called SCV phenotype [45,57–59]. SCVs exhibit slow growth and metabolic alterations [60] and are often the result of adaption to an intracellular lifestyle [59]. SCVs form slow-growing colonies on agar plates and lack haemolysis and pigmentation [60] however, JE2 *brnQ1* recovered from macrophages after 28 days of infection exhibited normal colony size as compared to the control (i.e. *brnQ1* bacteria grown in TSB), displayed haemolysis, and were pigmented (Fig 5g) indicating JE2 *brnQ1* recovered from macrophages are not SCVs.

In summary, growth of *S. aureus* within macrophages requires high affinity transport of BCAAs. BCAA deprivation, for instance mediated through *brnQ1* inactivation, results in *S. aureus* entering a state of *quiescence* that is distinct from SCV phenotypes. In the BCAA-starved state, *S. aureus brnQ1* stay viable within intact macrophage vacuoles for inordinate duration, even in the absence of antimicrobials.

## Discussion

At the forefront of host-pathogen interaction is a battle for nutrients (reviewed in [10,61,62]). The host deploys a plethora of effectors to deprive invading bacteria of essential nutrients whereas the ability of bacteria to overcome nutrient restriction underpins their successful colonization and infection. In this study we demonstrate that the availability of BCAAs and their transport by *S. aureus* are pivotal to the ability of the pathogen to grow inside macrophages.

Here, we found that inactivation of BrnQ1, the main BCAA transporter of *S. aureus,* is sufficient to arrest intracellular growth within the macrophage phagolysosome. Depriving *S. aureus* of BCAAs through a *brnQ1* mutation dramatically alters their physiology within the phagosome, resulting in major defects in gene expression and *de novo* protein synthesis. Consequently, *S. aureus brnQ1* mutants do not initiate appropriate patterns of gene expression and thus fail to adapt to the intracellular environment. This is emphasized by the inability of *brnQ1* bacteria to upregulate the expression of Fur-regulated genes which starkly contrasts wild-type bacteria that appropriately respond to iron limitation after phagocytosis by macrophages (Fig 2). Our data suggest that *S. aureus*, when deprived of Val and Leu, enter a state of *quiescence* within macrophages but can be "re-activated" once BCAAs become available.

The *S. aureus* genome encodes a complete BCAA biosynthesis machinery; however, the bacterium prioritises BCAA scavenging over *de novo* synthesis ostensibly because BCAA synthesis is energetically costly [21,28]. *S. aureus* possesses three functional BCAA transporters: BrnQ1 and BcaP, which import all three BCAAs, and BrnQ2, a designated Ile transporter [24,29]. A fourth gene encoding BrnQ3 bears homology to a BCAA transporter but lacks BCAA transport function [24]. The genes encoding the *S. aureus* BCAA transporters belong to the CodY regulon [25] and, as such, their transcription is de-repressed upon CodY inactivation [27,28]. Interestingly, there is a hierarchy of de-repression that is affected through CodY: modest decrease in intracellular BCAAs causes de-repression of *brnQ1*, whereas BCAA biosynthesis, which is also CodY regulated, remains repressed at the same level of CodY inactivation [28,63]. This is supported by our data showing that inactivation of *brnQ1* renders *S. aureus* unable to grow inside macrophages despite possessing the capability to synthesize the BCAAs required for growth. In this scenario, *brnQ1* must be expressed to some level by wild-type bacteria while the BCAA biosynthesis genes encoded by the *ilv* locus are likely not expressed.

In *S. aureus,* BrnQ1 is the preferred transporter for BCAA uptake [29]. For instance, when grown in CDM lacking Val or Leu (<34 µM), BrnQ1-mediated BCAA uptake is required for growth [24]. Although BcaP can also import all three BCAAs and exhibits similar substrate affinity and transport velocities, BrnQ1 remains the predominant transporter likely due to its higher level of expression as compared to BcaP [29]. Our experiments support this, as the provision of *bcaP* on a multi-copy plasmid is sufficient to rescue the growth of *brnQ1* bacteria inside macrophages. This indicates that BCAAs are still available to some extent inside the phagosome lumen, but clearly *brnQ1*-deficient bacteria cannot acquire sufficient Val and Leu to support growth. This suggests that *bcaP* is either poorly or not at all expressed in *S. aureus brnQ1* mutants. However, our transcriptome analysis did not show differential *bcaP* expression between JE2 *brnQ1* mutant and JE2 WT (Table 1).

Since *S. aureus brnQ1* mutants fail to replicate inside macrophages, but do not exhibit growth defects in complex media (TSB or CDM with tryptone), the intraphagosomal milieu must be BCAA-deplete although the exact concentrations are unknown. SLC38A9, a human lysosomal membrane transporter, has been shown to remove Leu from lysosomes and it may contribute to BCAA limitation [64]. Val must also be limited in the phagosome. This notion is supported by the fact that simultaneous Leu and Val supplementation is required to rescue the *brnQ1* mutant in macrophages (S13 Fig). It is unlikely that the phagosome is depleted of Ile, because the absence of this BCAA would lead to de-repression of CodY targets and subsequent *de novo* BCAA biosynthesis [21], followed by bacterial growth.

Our data show that inactivation of *codY*, and therefore de-repression of the CodY regulon, is sufficient to allow the growth of *brnQ1*-deficient *S. aureus* within macrophages. More specifically, our data reveal that CodY inactivation enables growth of *S. aureus brnQ1* mutants in an IlvD- and BcaP-dependent manner, indicating that BCAA biosynthesis and transport are enhanced in the *codY* background. This raises an important question: why wouldn't the genes that allow growth simply be de-repressed in *brnQ1*-deficient *S. aureus*? One possible explanation is the complexity of CodY regulation, which is influenced by intracellular Ile availability and by GTP levels within the bacteria [20,21]. Moreover, specific gene targets can be regulated not only by CodY but also by other gene expression modulatory mechanisms. For instance, the *ilv* operon, which encodes BCAA biosynthesis, is not only subject to CodY regulation but is also influenced by the formation of a leucine-rich attenuator peptide that downregulates the expression of *ilv* genes [21]. This peptide is encoded by an RNA, which was termed Rli60 in *Listeria monocytogenes*, where its role in fine-tuning the CodY response has been demonstrated [65].

Previous studies had concluded that *S. aureus* is auxotrophic for BCAAs Leu and Val [66] because the bacteria delay induction of BCAA synthesis in environments with low BCAA levels [24], especially if Ile is present as a co-repressor with CodY. Macrophage cultivation and infection in the presence of high BCAA concentrations is sufficient to rescue the replication defect displayed by *brnQ1* mutants, which highlights the importance of these metabolites for *S. aureus* pathogenicity. Several human pathogens require functional BCAA transport or biosynthesis for full virulence (reviewed in [23]), and our study reaffirms the pivotal role of these metabolites at the host - pathogen interface.

Mature phagolysosomes are highly degradative organelles, in part, due to their acidic pH and enrichment with proteolytic enzymes such as cathepsins that degrade proteins thereby liberating peptides [67]. In addition, *S. aureus* secretes several proteases (e.g., aureolysin or staphopain) that can degrade various substrates and liberate peptides that could serve as nutritional source [68]. Indeed, the *S. aureus*-containing phagosome has been shown to be proteolytic [33]. Through the activity of the oligopeptide permease Opp3 [68] and the di-tri-peptide permease DtpT [69], *S. aureus* could, at least in theory, utilize peptides to satisfy its amino acid requirements under conditions of nutrient limitation. Nevertheless, we found that inactivation of *brnQ1* alone is sufficient to inhibit *S. aureus* growth inside the macrophage phagosome. Although protease expression is enhanced upon *codY* inactivation [25,70], we find that simultaneous mutation of *brnQ1, bcaP* and *ilvD* is sufficient to render a *S. aureus codY* mutant unable to grow intracellularly. Our transcriptomic analysis revealed that genes in the *opp3* operon were some of the few loci upregulated by intracellular *S. aureus brnQ1* mutants suggesting that the bacteria may utilize this peptide transporter to maintain viability. The fact that *brnQ1*- or *brnQ1/bcaP/codY/ilvD*-deficient *S. aureus* still fail to grow inside macrophages, would indicate that peptides are either not being used by the bacteria or that these peptides do not provide enough BCAAs to support growth. Notably, all strains with a *codY* mutation display increased proteolytic activity *in vitro*. Still, *S. aureus codY* mutants which additionally lack *brnQ1, bcaP* and *ilvD* (quadruple mutants), remain unable to acquire sufficient BCAAs to support growth withing macrophages (S16 Fig), indicating that peptides do not provide enough BCAAs to overcome macrophage restriction. These observations provide important insight into the contribution of free amino acids obtained from the extracellular environment versus protein degradation and peptides within the phagosome as a source of nutrition which allows *S. aureus* replication within the phagosome lumen.

The impact of *brnQ1* inactivation on the physiology of intracellular *S. aureus* is striking and forces the bacteria into a non-replicative state, causing them to fail to regulate adaptiveness to environmental cues. Decreased proliferation rates and/ or growth arrest and metabolic quiescence, accompanied by antibiotic tolerance, are hallmarks of *persistence* [50–52]. So-called *persisters* can arise stochastically within a bacterial population [50], or in response to environmental cues, such as exposure to antibiotics [43], nutrient starvation, oxidative stress [54], or internalization by host cells, particularly macrophages [53,71]. Bacterial persisters are thought to constitute a reservoir for relapsing infections, associated with therapeutic failures [43,71].

Various mechanisms were shown to trigger *S. aureus persisters*, such as antibiotic exposure of intracellular *S. aureus* [43,54]. In macrophages, for example, stringent response induction [43] and ROS-induced defects in translation and ATP synthesis were proposed as mechanisms of persistence [54,55]. Moreover, ROS was shown to mediate intracellular *S. aureus* antibiotic tolerance by disrupting the TCA cycle [55].

Antibiotic pressure is a major factor in triggering intracellular persistence of *S. aureus* [43,54,55,72]. SCVs, an extensively studied form of *S. aureus* persisters [51,57,60,73–76], may arise in the absence of antibiotic selective pressure [77]. They comprise a slow-growing subpopulation of bacteria with very distinctive phenotypic characteristics: reduced haemolysis and cytotoxicity, variations in pigmentation as compared to parental strains, enhanced antibiotic tolerance and slower growth rates, often due to auxotrophies (e.g., menadione or haemin) [78], resulting in non-haemolytic pinpoint-sized colonies on agar plates.

*S. aureus brnQ1* mutants recovered here from macrophages, however, despite being growth-arrested and non-responsive to environmental cues (i.e., iron depletion; aTc-induction of the *Cerulean* fluorescent protein), do not exhibit SCV characteristics, nor do they display other phenotypes reported for persisters (e.g., induction of stringent response [43]).

The phenotype reported here, conforms to the definition of *dormancy,* as stated by Balaban et al.[51]: "*Dormancy reflects the state of a bacterium that does not grow and has decreased activity when compared with growing cells*". This phenotype appears to be a non-antibiotic-driven form of intracellular dormancy triggered by BCAA starvation within macrophage phagolysosomes. This dormancy is reversible, as evidenced by the sporadic outgrowth of the bacteria from infected macrophages, in what appears to be, a stochastic event.

The environmental cues or genetic changes that enable this process are currently unknown and are of research interest. One possible explanation is that the bacteria inactivate CodY, however re-infection of macrophages with *brnQ1* bacteria that displayed *URG* does not alter or enhance the course of a subsequent infection. Furthermore, genome sequence analysis revealed no SNPs in *codY* or *ilv* genes, which were previously reported to be "hot spots" for the accumulation of suppressor mutations that allow *S. aureus* to eventually replicate under BCAA-deprived conditions [21]. Further investigation into how *S. aureus brnQ1* exits *dormancy* is ongoing and will provide insight into the mechanisms *S. aureus* employs to modulate its metabolic program for growth and survival under nutrient-limited conditions.

In summary, our study revealed a pivotal role for BCAA transport in the growth and survival of *S. aureus* inside the macrophage phagolysosome. Remarkably, inactivation of BrnQ1 alone is sufficient to arrest intracellular growth of *S. aureus* and force the bacteria into a state of temporary *dormancy.* These growth-arrested *S. aureus* reside within viable phagocytes, in the absence of any antibiotic pressure, and could serve as a long-term reservoir of bacteria with complete virulence potential, should BCAA starvation be alleviated.

Understanding the mechanisms underlying this state of dormancy and how the bacteria "reactivate" is important in the rational development anti-staphylococcal therapies especially since bacterial amino acid and peptide transporters are emerging as an attractive target in the fight against virulent or antibiotic-resistant *S. aureus* [79]. However, our observations suggest that anti-staphylococcal strategies targeting bacterial amino acid transporters such as BrnQ1 should be carefully tailored to avoid selection of dormant intracellular populations prone to long-term intracellular survival and potential dispersal to secondary infection sites.

## Materials and Methods

### Ethics Statement

Leukocyte samples (i.e., leukoreduction system chambers from anonymous healthy donors) were obtained from the Institute for Clinical Transfusion Medicine and Haemotherapy, of the University of Würzburg, Germany for research purposes and require no specific approval by an ethics committee. Whole human blood was also drawn from healthy adult volunteers upon verbal consent in accordance with protocols approved by the University of Western Ontario's Research Ethics Board, London, Ontario, Canada.

### Bacterial culture conditions

Bacterial strains are listed in S1 Table. All liquid cultures were grown aerobically at 37°C with 200 rpm shaking. *E. coli* was grown in Luria-Bertani broth (LB) or on LB agar containing 100 µg/mL ampicillin for plasmid maintenance when necessary. *S. aureus* strains were routinely grown in tryptic soy broth (TSB), or on TSB agar plates (TSA) if not stated otherwise. Media were supplemented with appropriate antibiotics, when necessary: 10 µg/ml chloramphenicol (plasmid encoded antibiotic resistance); and/or 5 µg/ml erythromycin, 5 µg/mL tetracycline or 50 µg/mL kanamycin (for chromosomally encoded antibiotic resistance).

### Construction of bacterial strains and plasmids

All used strains (S1 Table), plasmids (S2 Table) and oligonucleotides (S3 Table) can be found in the Supporting material. Strains used for macrophage infection experiments shown in Fig 1c were retrieved from the Nebraska Transposon mutant library [34].

The *S. aureus* insertional transposon mutant of *brnQ1* (NE945) was retrieved from the Nebraska Transposon mutant library, transposon insertion at the correct position was confirmed by PCR, using primers brnQ1_fwd and brnQ1_rev and the mutation was further transduced via phage φ11 or φ80 into the genetic background of wild-type *S. aureus* strains to yield JE2 *brnQ1*, 6850 *brnQ1*, SH1000 *brnQ1* and RN4220 *brnQ1* respectively.

Plasmid p*brnQ1* was generated to complement the *brnQ1* gene *in trans* under the transcriptional control of its native promoter, using Fast Cloning [80]. Briefly, the open reading frame of *brnQ1* gene (locus ID: SAUSA300_0188/ SAUSA300_RS00985) was amplified from *S. aureus* JE2 genomic DNA together with its native promoter region included in the 400 bp upstream of the start codon, using primer pairs brnQ1_Comp_Inf_fwd and brnQ1_Comp_Inf_rev. The pALC2085 vector [81] was linearized by PCR using primer pairs p2085_Inf_for brnQ1_fwd and p2085_Inf_for brnQ1_rev. Both insert and vector were amplified using an 18-cycle amplification protocol and Phusion DNA-polymerase (Thermo Fisher Scientific, Cat. No. F530L). 1 µL DpnI enzyme (Thermo Fisher Scientific, Cat. No. ER1701) was added to each 50 µL un-purified PCR reaction, vector and insert were mixed in a 1:1 (v/v) ratio and DpnI digest was carried out for 1h at 37°C. 2 µL of the reaction was directly transformed into electrocompetent *E. coli* DH5α and plated on LB agar plates supplemented with 100 µg/mL ampicillin. The plasmid p*brnQ1* was isolated from *E. coli* and the sequence was confirmed by Sanger sequencing (SeqLab, Göttingen). Subsequently, plasmids were electroporated into *S. aureus* RN4220 and further transduced into *S. aureus* JE2 using phage φ11. A similar procedure was used to generate a control, by amplifying a promoter-less *brnQ1* using primer pairs brnQ1_Comp_Inf_fwd and brnQ1_P-less_Comp_Inf_rev and cloning it into the pALC2085 backbone to yield pP$_{less}$_*brnQ1*. Plasmids p*bcaP* and p*brnQ1* shown in Fig 4d were previously described [24,29] and carry *bcaP* or *brnQ1,* respectively, that were cloned into the expression plasmid pRMC2 under the control of their respective native promoters.

Red-fluorescent or cyan-fluorescent expressing *S. aureus* JE2 were generated by transducing the pSarAP1-mRFP or p*Cerulean*, respectively, from recombinant *S. aureus* RN4220 strains using phage φ11.

*S. aureus* JE2 *fur* and JE2 *brnQ1/fur* were generated by transducing the *tet* cassette from *S. aureus* MJH010 *fur::tet* [40] using phage φ11.

Strains in USA300 carrying either *codY* (locus ID: SAUSA300_1148/ SAUSA300_RS06210) or *ilvD* (locus ID: SAUSA300_2006/ SAUSA300_RS11035) mutations were created by transducing either *codY*::Tn (NE1555) or *ilvD*::Tn-Kan$^R$ using phage φ80 into the recipient strain.

The replacement of the erythromycin resistant cassette in strain JE2 NE718 (*ilvD*) was done using the plasmid pKAN as previously described [82]. In brief, pKAN plasmid was transduced into JE2 NE718 (*ilvD)* using phage φ80. Replacement of the Ery$^R$ cassette for Kan$^R$ was confirmed by resistance phenotype (i.e., the strain became Ery$^R$ and Kan$^R$) and by PCR analysis.

## Isolation of primary human monocytes from human peripheral blood

Primary human macrophages were derived from peripheral blood mononuclear cells (PBMCs) isolated from whole blood leukoreduction system (LRS) chambers [83] using the SepMate-50 system (StemCell Technologies, Cat. No. 85450) and Ficoll-Paque PLUS density gradient medium (Sigma Aldrich, Cat. No. GE17-1440-03), according to manufacturer's instructions. Monocytes were then purified from the PBMC fraction using the EasySep Human CD14 Positive Selection Kit II (StemCell Technologies, Cat. No. 17858) according to the manufacturer's instructions. In some instances PBMCs were isolated from whole human blood using lympholyte-poly (Cedarlane labs) according to the manufacturer's instructions and as described [39].

## Macrophage differentiation and polarization

All macrophage cultivation steps, and infection experiments were done in RPMI1640 (Thermo Fisher Scientific, Cat. No. 72400054) supplemented with FBS (10% v/v; heat inactivated: 56°, 30 min) henceforth termed *Infection Medium,* unless otherwise indicated. Purified monocytes were seeded in Nunc UpCell cell culture plates (Thermo Fisher Scientific, Cat. No. 174902) at a density of 5-7x $10^6$ cells/plate in *Infection Medium* supplemented with 50 ng/mL macrophage-colony-stimulating factor (M-CSF; StemCell Technologies, Cat. No. 78057) and incubated at 37°C in a humidified incubator with 5% $CO_2$ for 7 days. Following the 7-day differentiation protocol, macrophages were detached from the Nunc UpCell by incubation at 20°C, following the manufacturer's instructions. Cells were collected, centrifuged, re-suspended in *Infection Medium*, counted using a Neubauer counting chamber and seeded in cell culture well plates. In some instances isolated PBMCs were adhered directly to 12 well tissue culture treated plates and differentiated in the presence of recombinant human M-CSF at 10 ng/mL (PeproTech, Cat. No. 300–25) as previously described [39]. Differentiated macrophages were used for infections from day 7 to day 9.

## Macrophage infection assay

For standard infection experiments, $10^5$ cells were seeded into 24 well microtiter plates, 24h prior to infection. Bacterial overnight cultures (TSB) were diluted to an $OD_{600}$ = 0.5 and grown in TSB for 45 min at 37 °C, under continuous shaking at 200 rpm. Bacteria were washed with Dulbecco's phosphate-buffered saline (DPBS, Thermo Fisher Scientific, Cat. No. 14190169), diluted in *Infection Medium* and counted using a Thoma counting chamber (Hartenstein, Germany, Cat. No. ZK04). For counting, 10 µL of the above-described suspension were loaded onto the Thoma chamber (*small* counting square area: 0.0025 mm$^2$; depth 0.1 mm). Bacteria were allowed to sediment for 10 min, then counted in at least 16 *small* squares and the "chamber factor" of 4 x $10^6$ was used to estimate no. of bacteria/mL. Non-opsonized bacteria were used to infect macrophages at a multiplicity of infection (MOI) of 5. For the synchronization of infection, plates were centrifuged at 200 x g, for 5 min, without brakes and then incubated for 30 min at 37°C in an incubator with humidified atmosphere with 5% $CO_2$. After 30 min, extracellular bacteria were removed by 15 min incubation in *Infection Medium* containing 30 µg/ml lysostaphin (AMBICIN L, AMBI), followed by washing twice with sterile DPBS and further incubation in *Infection Medium* without antibiotics until the end of the experiment. For long-term experiments, viable infected macrophages were cultivated for up to 28 days in the absence of lysostaphin (except for the initial 15 min pulse).

In some instances, primary human and RAW 264.7 macrophage infections were performed using 12 well tissue culture treated dishes as previously described [12]. Here macrophages were infected at an MOI of 5 in serum-free RPMI1640. After synchronization by centrifugation at 277 x g for 2 min, phagocytosis was allowed to occur for 30 min. At this time the medium was replaced with serum-free RPMI1640 containing 100 µg/mL gentamicin for 1 hr, to kill the extracellular bacteria, followed by washing twice with sterile DPBS and further incubated with *Infection Medium.* For RAW 264.7 macrophage experiments, *Infection Medium* with 5% (v/v) FBS was used instead.

### Intracellular bacterial replication assay (CFU)

To enumerate intracellular bacteria, at the required time points, cells were washed once with DPBS and lysed using alkaline water (pH11) for 5 min at room temperature. Dilution series of the complete cell lysate were plated on TSA and recovered bacterial colony forming units (CFU) were counted. "T0" represented in all graphs corresponds to 45 min *p.i.* (i.e., 30 min phagocytosis and 15 min lysostaphin treatment). In some instances, after gentamicin treatment and DPBS washing, infected macrophages were lysed using 0.5 mL of 0.1% (v/v) Triton X-100 diluted in DPBS. Lysis at this time equals 1.5h *p.i.* and the CFU/mL represents the number of phagocytosed or gentamicin protected bacteria. At 20h *p.i.* supernatants from all wells were collected individually and centrifuged to pellet any bacteria that may have emerged from infected macrophages. While centrifuging, wells containing macrophages were lysed using Triton X-100 as described immediately above, and this lysate was used to resuspend any pelleted bacteria. At each time point bacteria were serially diluted onto TSA and the CFU determined. To calculate the fold change at 20h *p.i.,* the CFU/mL at 20 h was divided by CFU/mL at 1.5h. The data were expressed as the Log10 fold change.

### Exogenous BCAA complementation

Macrophages were seeded 24h prior to infection in either *Infection Medium* or *Infection medium* supplemented with 1 mM of each branched-chain amino acid (Ile, Leu, Val) and all infection steps were carried out in the respective medium. Samples were further processed for CFU counting or RNA extraction at 45 min ("T0"), 10h and 24h *p.i.* (CFU) or 10h *p.i.* (RNA). In some instances, 1 mM of either Val, Leu or both were added to RPMI1640 + 5% FBS that was added back to RAW 264.7 macrophages immediately after gentamicin treatment. CFU determination was performed as described above.

### Live Cell Imaging

4.5 x $10^4$ macrophages/well were seeded in 8 well chamber µ-slides (ibidi, Cat. No. 80826) 24h prior to infection. Infection with mRFP-expressing bacteria was performed at an MOI of 5 as described above. Activation of effector caspases 3/7 in infected macrophages was visualised 5h *p.i.* by the addition of 5 µM of the fluorogenic substrate Cell Event Caspase-3/7 Green Detection Reagent (Thermo Fisher Scientific, Cat. No. C10723).Time-lapse imaging was performed in *Imaging Medium* (*Infection Medium* without phenol red), in the absence of antibiotics, on a Leica TCS SP5 confocal laser scanning microscope (Leica GmbH, Wetzlar, Germany; software LAS AF version 2.7.3.9723) using a 40x oil immersion objective (Leica HC PL APO, NA = 1.3), equipped with a pre-warmed (37°C) live-cell incubation chamber (Life Imaging Systems) and perfused with a humidified atmosphere containing 5% $CO_2$. The LAS AF software was used for setting adjustment and image acquisition. All images were acquired at a resolution of 1024x1024 and recorded in 8-bit mode, at 30 min time intervals. Z-stacks were imaged with a step size of 0.4 µm. Image-processing was performed using Fiji [84].

### Phase contrast microscopy

Phase contrast microscopic images of live, infected cells were acquired with a LEICA DM IRB microscope connected to a SPOT Pursuit USB camera using a Leica C Plan L 20 x/0.30 PH1 objective and VisiView software (Visitron).

## Transmission Electron Microscopy (TEM)

TEM sample preparation and imaging were carried at the Imaging Core Facility of the Biocenter, University of Würzburg, Germany. Macrophages were seeded on cover slips and infection was performed as described. Infected samples were washed once in DPBS, fixed with 2.5% glutaraldehyde (in 50 mM sodium cacodylate [pH 7.2], 50 mM KCl and 2.5 mM MgCl$_2$) for 2h at 4°C. Samples were subsequently washed twice for 5 min with 50 mM cacodylate buffer (pH 7.2) and were further fixed with 2% buffered OsO$_4$ (Science Services) and contrasted with 0.5% uranyl acetate in distilled water and finally embedded in Epon (Serva) after ethanol-based dehydration. Ultrathin sections were generated from embedded samples and contrasted with 2% uranyl acetate in ethanol and lead citrate. TEM imaging was performed on a JEOL JEM-2100 instrument.

## Preparation of cells for RNA extraction

For total RNA extraction (host and pathogen), 3 x 10$^6$ macrophages grown in 6-well cell culture plates (i.e., 1.5 x 10$^6$ macrophages were seeded per well and 2 wells/condition were pooled) were infected at MOI of 5, as described. At 10h *p.i.*, cells were washed once with DPBS, and RNAprotect Cell Reagent (Qiagen, Cat. No. 76526) was added immediately (1 part *Infection Medium* and 5 parts RNAprotect). Samples were stored at 4°C for a maximum of one week before further processing.

Experiments with bacteria grown in *Infection Medium* were carried out in parallel, using the same bacterial inoculum. Bacteria were similarly grown in 6-well plates at 37°C in a humidified incubator with 5% CO$_2$, without shaking. After 10h, bacteria were collected by centrifugation and pellets were flash-frozen in liquid N$_2$ and stored at -80°C.

For experiments depicted in Fig 3a, bacteria were grown overnight in TSB, washed 2x in DPBS and resuspended in *Infection Medium* at OD$_{600}$ = 1. Bacteria were grown for 1h at 37°C with 200 rpm shaking, collected and flash-frozen in liquid N$_2$ and stored at -80°C.

## RNA extraction

RNA extraction from either *S. aureus*-infected macrophages or cultures of *S. aureus* was done using the RNeasy Mini Kit (Qiagen, Cat. No. 74104). Infected cells stored in RNAprotect Cell Reagent were collected in RNase-free tubes, by centrifugation (6,200 x g, 5 min, 4°C). Frozen bacterial pellets were allowed to thaw on ice for 5 min. Next, 600 µL RLT Buffer (kit) supplemented with β-mercaptoethanol (10 µL) was added to the pellets, samples were disrupted in a FastPrep FP120 instrument (6 m/s for 45 sec, at 4°C) using Lysing Matrix B tubes (MP Biomedicals, Cat. No. 116911100), immediately cooled down and centrifuged (10,000 x g, 2 min, 4°C). Following lysis, manufacturer's instructions for the RNeasy Mini kit were followed. TURBO DNA-*free* DNase (Thermo Fisher Scientific, Cat. No. AM1907) was used to digest contaminating DNA (1h, 37°), according to manufacturer's instructions. For routine applications, concentration and purity were determined using a Nanodrop instrument and non-denaturing agarose gel electrophoresis, using ethidium bromide staining [85]. The absence of DNA contamination was verified by qPCR (40 cycles using *gyrB*-specific primers and RNA samples as template).

## RNA sequencing

RNA-seq of *S. aureus* grown in *Infection Medium* as well as dual RNAseq of *S. aureus*-infected macrophages were conducted by the Core Unit SysMed at the University of Würzburg (Germany). RNA quality was checked using a 2100 Bioanalyzer with the RNA 6000 Nano kit (Agilent Technologies, Cat. No. 5067–1511). Ribosomal RNA (rRNA) was depleted from 200 ng of total RNA as input using Lexogen's RiboCop rRNA Depletion Kit protocol according to manufacturer's recommendation with the following modifications: for mixed human and bacterial samples, 5 µl of RiboCOP V1.2 probes were mixed with 5 µL of RiboCOP META probes, 4 µl hybridization buffer, and 21 µL of RNA sample. Following rRNA depletion, DNA libraries suitable for sequencing were prepared using CORALL Total RNA-Seq Library Prep protocol according

to manufacturer's recommendation with 14 PCR cycles. After quality control, libraries were sequenced on an *Illumina NextSeq* platform (1*75 bp single end run). Demultiplexed FASTQ files were generated with bcl2fastq2 v2.20.0.422.

Sequencing reads were trimmed for Illumina adapter sequences using Cutadapt version 2.5 with default parameters. In addition, *Cutadapt* was given the *–nextseq-trim = 20* switch to handle two colour sequencing chemistry and reads that were trimmed to length 0 were discarded. Reads were filtered with *FastQScreen* [www.bioinformatics.babraham.ac.uk/projects/fastq_screen/]. The splice-aware aligner STAR [86] was used for the mapping of reads originating from human cells to the human genome (GRCh38.p13), whereas the Reademption pipeline [87] was used to map reads originating from *S. aureus* to the reference (GCF_000013465.1_ASM1346v1). Read counts for each gene were generated and the count output was utilized to identify differentially expressed genes using DESeq2 v1.24.0 [88].

### RT-qPCR

1 µg total RNA was converted to cDNA using the RevertAid First Strand cDNA Synthesis kit (Thermo Fisher Scientific, Cat. No. K1622) according to manufacturer's instructions, using random hexamer primers. RT-qPCR was performed one a StepOne Plus instrument (Applied Biosystems). Each reaction mix consisted of GreenMasterMix (Genaxxon, Cat. No. M3052.0500), 900 nM of each primer and 50 ng (infected samples) or 10 ng (bacteria alone) cDNA. Relative gene expression was normalized to expression of the *gyrB* reference gene and represented as -ΔCt values [89]. Oligonucleotides used in RT-qPCR experiments are listed in S3 Table.

### Genome sequencing

DNA was isolated using the Promega One4All kit on a KingFisher (ThermoFisher Scientific) device. DNA concentration was measured using Qubit (dsDNA HS assay kit, Thermo Fisher Scientific, Germany) and sequencing libraries were generated using the Nextera XT library Prep Kit (Illumina, USA). Data acquisition was performed on a NextSeq550Dx (Illumina, USA) during a 2 × 150 bp paired-end sequencing run using a mid-output cassette. The derived FastQ files were quality-controlled, and the genomes were assembled using SKESA (V2.4.0 on Seqsphere+ (client and server version 9.0.0, Ridom GmbH, Münster). Contigs were aligned and reordered using progressiveMauve [90] and the genomes were annotated using prokka [91].

### *Cerulean* induction in infected macrophages

Macrophages were seeded on coverslips and infected as described with JE2 WT or JE2 *brnQ1* carrying p*Cerulean* (anhydrous tetracycline (aTc)-inducible expression of the codon-adapted fluorescent protein Cerulean [81]). At either 1h or 10h *p.i. Cerulean* was induced by the addition of 300 ng/mL aTc, and samples were incubated for further 2h before fixing with 4% paraformaldehyde (PFA, Morphisto, Cat. No. 11762.01000). Fixed cells were washed 3x in DPBS, permeabilized with 0.2% Triton-X100 (in DPBS) for 5 min, followed by incubation with 50 mM ammonium chloride for 5 min. Staining was done for 20 min, with a solution containing 1 µg/mL BODIPY FL Vancomycin (Thermo Fisher Scientific, Cat. No. V34850), Phalloidin-iFluor647 (Abcam, Cat. No. 176759; diluted according to manufacturer's protocol) and 4 µg/mL Hoechst 34580 (Thermo Fisher Scientific, Cat. No. H21486), in DPBS. Cover-slips were mounted on microscopy slides with Mowiol and air-dried overnight.

Image acquisition was performed on a Leica TCS SP5 CLSM using a 40x oil immersion objective (NA 1.4). Per sample at least 10 fields of view were recorded with each covering an area of 388 × 388 µm. Images were recorded at a resolution of 1024 x 1024 pixels at 8-bit grey scale depth.

Imaging was done using the following excitation/emission (Ex/Em) settings: Cerulean: Ex/Em = 458/460–494 nm (Argon laser); BODIPY FL Vancomycin: Ex/Em = 488/499–586 nm; (Argon laser); Hoechst 34580: Ex/Em = 405/423–477 nm (diode laser); Phalloidin-iFluor 647 Ex/Em = 633/634–710 nm (HeNe laser).

Image analysis was conducted using Fiji [84], using an in-house macro toolset, which uses BODIPY-FL- channel to identify the macrophage-associated bacteria, whose positions are saved as individual regions-of-interest (ROI). Subsequently, the mean fluorescence intensities in each ROI were determined for both the Cerulean channel and the BODIPY-FL channel. The results were exported from Fiji as text files with comma-separated values (csv). The csv-files fere transformed into text-format using RStudio (Integrated Development for R. RStudio, PBC, Boston, MA) and subsequently imported the data as dot plots into the flow cytometry software Flowing2 (Turku Bioscience Centre, Turku Finland) to determine percent of total bacteria expressing *Cerulean*.

**Fluorescence-based proliferation assay**

Fluorescence proliferation assays were performed as previously described [18,92] using GFP expressing *S. aureus* USA300 and Δ*brnQ1*. In brief, bacteria were labelled with Cell Proliferation Dye eFluor 670 |(eBioscience, Cat. No. 65-0840-85) and used to infect macrophages that were grown on 18 mm glass coverslips (No. 1) at an MOI of 5 as described above. At 1.5, 10 and 20h *p.i.* macrophages were stained with 1 µg/mL tetramethylrhodamine-conjugated wheat germ agglutinin (TMR-WGA, Thermo Fischer Scientific, Cat. No. W849). Stained cells were fixed with 4% (v/v) PFA at room temperature for 20 min and mounted on glass microscope slides using ProLong Gold Antifade Mountant (Thermo Fischer Scientific, Cat. No. P36934). Confocal fluorescence microscopy was performed using a Zeiss LSM 880 with Fast AiryScan equipped with a Plan-Apochromat ×63/1.4 oil DIC M27 objective and Definite Focus 2. GFP, TMR-WGA and eFluor 670 were excited with available laser lines at 488 nm, 561 nm, and 633 nm, respectively. Post-image processing (i.e., contrast enhancement and image cropping) was conducted using Fiji [84].

**Bacterial colony morphology analysis**

*S. aureus* JE2 *brnQ1* were recovered from infected macrophages by lysis with pH11 water. Colony pigmentation was assessed by plating the lysate onto TSA plates. Haemolysis was assessed by plating lysates onto Columbia agar with 5% defibrinated sheep blood. After 30h incubation at 37°C, plates were photographed and colony diameter was measured with Fiji [84]. Fresh bacteria grown in late-exponential phase in TSB served as control.

**Bacterial growth curves**

Bacterial growth curves were recorded in 48-well plates using a TECAN MPlex plate reader. *S. aureus* strains were grown overnight in the appropriate culture medium: TSB, RPMI1640 or *Infection Medium*. Next day, cultures were diluted to an $OD_{600} = 0.1$ in fresh corresponding medium and used to inoculate 400 µL/well. Sterile medium was used as reference. For experiments depicted in S1 and S13a Figs, bacteria were grown overnight in TSB and washed 4x in DPBS before inoculation at $OD_{600} = 0.1$, in a chemically defined medium (CDM) [[93] with 25 mM glucose]. BCAAs were excluded from the CDM formulation, as needed. Absorbance at 600 nm ($OD_{600}$) was measured every 18 min for 20h or 50h, under continuous shaking at 200 rpm. For experiments depicted in S1 Fig, bacterial doubling times were estimated from growth curves.

**Bacterial growth in valine-free CDM**

For experiments depicted in S14 Fig, CDM was prepared as previously described [24] and Val was omitted when needed. Bacteria were grown overnight in TSB, pelleted and washed in 1 mL of sterile saline (0.9% w/v NaCl). Bacteria were diluted into saline at an $OD_{600}$nm of 0.1 and 10 µL of this suspension was used to inoculate 2mL culture volumes to give a starting $OD_{600}$nm of ~ 0.0005. The bacteria were grown in 13 mL polypropylene snap cap tubes at 37°C with constant shaking at 250 rpm. The end point $OD_{600}$ was measured at 24h. As controls, either 1 mM Val or 0.1% (w/v) tryptone was added to the culture medium.

### S. aureus infection of non-professional phagocytic cells

HeLa cells (HeLa 229, ATCC CCL-2.1) and 16HBE14o- human bronchial epithelial cells were routinely cultured in *Infection Medium*, at 37 °C and 5% $CO_2$. For infection, $10^5$ cells/well were seeded in 12-well microtiter plates 24h prior to infection. Infections were carried out with a multiplicity of infection (MOI) of 5 as described for macrophage infections. In this case, however, bacteria and host cells were co-cultivated for 1h, after which extracellular bacteria were removed by a 30 min treatment with 20 µg/ml lysostaphin (AMBICIN L, AMBI) followed by further incubation in medium containing 2 µg/ml lysostaphin until the end of the experiment.

### Phagosomal escape assay

Phagosomal escape was determined as previously described [94]. Briefly, HeLa YFP-CWT or 16HBE14o- YFP-CWT cells were infected with *S. aureus* at MOI of 5 in 24 well µ-plates (ibidi, Cat. No. 82406). For experiments depicted in S4b, S4c and S5b Figs, mRFP-expressing bacterial strains were used. For experiments shown in S5c Fig, bacteria were stained with SNARF-1 (Thermo Fisher Scientific, Cat. No. S22801) prior to infection. For this, logarithmic growth-phase bacteria grown in TSB were washed once with DPBS and stained with 8 µM SNARF-1 in a final volume of 500 µL DPBS, for 20 min, at RT, protected from light. Afterwards, samples were washed twice in DPBS and de-stained in fresh DPBS for 15 min, at 37°C, protected from light. After de-staining, samples were again washed twice in DPBS, re-suspended in *Infection Medium*, counted using a Thoma chamber as described, and used for infection. To ensure that the SNARF-1 staining does not interfere with bacterial growth, growth curves using either SNARF-1 stained or unstained bacteria were performed in both TSB medium and RPMI1640. Infection was performed as described. At 3, 6 and 8h post infection, cells were washed, fixed with 4% PFA overnight at 4°C, treated with 50mM $NH_4Cl$ in DPBS for 5 min, permeabilized with 0.1% (v/v) Triton X-100 and nuclei were stained with 4 µg/mL Hoechst 34580. Images were acquired with an Operetta automated fluorescence microscopy system (Perkin Elmer Operetta High-Content Imaging System) and analysed with the built-in Harmony Software. Phagosomal escape was indicated by the co-localization of the YFP-CWT (*phagosomal escape*) and mRFP/SNARF signals (*total intracellular bacteria*).

Live cell imaging experiments shown in S4 Fig and S2–S5 Movies were performed as described for macrophages, using mRFP-expressing bacteria.

### Cytotoxicity assay (LDH release)

The cytotoxicity of intracellular bacterial strains against epithelial cells (S5e Fig) and the cytotoxicity of bacterial supernatants against HeLa cells (S6a Fig), were analysed by measuring the amount of released lactate dehydrogenase (LDH). At the required time points, spent medium of infected/treated cells (~2x$10^5$ cells/sample) was collected, centrifuged for 5 min at 100 x g and 100µl of supernatant (in technical duplicates) was used to determine LDH release using the Roche Cytotoxicity Detection Kit[PLUS] (Sigma Aldrich, Cat. No. 4744934001), according to manufacturer's instructions. Spent medium of uninfected cells served as negative control and lysed cells served as positive control.

### S. aureus culture supernatant treatment of epithelial cells

Bacteria were grown for 24h in TSB supplemented with the appropriate antibiotics, if necessary. Cultures were adjusted to an $OD_{600}=4$, in a final volume of 1mL, centrifuged at 20,000 x g and supernatants were filtered through 0.22 µm-membrane filters. 2 x $10^5$ seeded HeLa cells were treated with 10% (v/v) supernatants (in *Infection Medium*) and incubated for 4h at 37°C and 5% $CO_2$. Cytotoxicity was measured by quantifying LDH released into the supernatant. 10% sterile TSB medium was used as negative control.

## Iron quantification in bacterial pellets

A colorimetric Iron Assay kit (Sigma Aldrich Cat. No. MAK025) was used to measure bacteria-associated iron. Overnight *S. aureus* JE2 cultures in RPMI1640 were transferred to RPMI1640 supplemented with 20 µM $FeSO_4$ ($FeSO_4$ heptahydrate dissolved in technical grade $H_2SO_4$), at $OD_{600}$ = 0.1 and grown for 24h at 37°C, under shaking at 200 rpm. $OD_{600}$ = 0.5 equivalents of bacterial cultures were pelleted by centrifugation and lysed for 30 min at 37°C in 110 µL *Lysis Buffer* (200 µg/mL lysostaphin, 1.2% Triton-X100, in water). Lysed samples were mixed with 110 µL Assay Buffer (kit) and incubated for 10 min at 80°C. Samples were subsequently centrifuged (13,000 x g, 10 min, 4°C) and 100 µL supernatant was used to measure total iron ($Fe^{2+}$ and $Fe^{3+}$), according to manufacturer's instructions.

## *Cerulean* induction in bacterial cultures

Induction of the *Cerulean* fluorescent protein in *S. aureus* JE2 during growth in TSB medium was measured by monitoring fluorescence and $OD_{600}$, in a TECAN MPlex plate reader, in black 96 well plates (Thermo Fisher Scientific, Cat. No. 165305). Overnight bacterial cultures were diluted to $OD_{600}$ = 0.1 in fresh TSB, supplemented with 300, 150 ng/mL aTc or vehicle control (ethanol). Absorbance at 600 nm and fluorescence ($Ex/Em_{433/504\ nm}$) were measured every 18 min for 16h. Results are expressed as arbitrary fluorescence units (Cerulean/$OD_{600}$ ratio).

## Random unrestricted growth (*URG*)

Bacterial unrestricted growth (*URG*) observed in infected macrophages, was analysed as follows: Primary human macrophages were seeded in 96-well plates at a density of 1.5 x $10^4$ cells/well and infected with *S. aureus* WT or its isogenic *brnQ1* mutant as described. Macrophages originating from a single donor were employed for each experiment to exclude donor effects on experimental outcome. Plates were observed daily and the number of wells, as well as the time point when *URG* occurred were recorded and plotted in a Kaplan-Meier survival plot, where each well represented a test subject. Wild-type *S. aureus*-infected macrophages and uninfected macrophages were used as positive (*URG* at 24h) and negative controls (no *URG*), respectively.

## Assessment of secreted protein profiles of different *S. aureus* strains

Single bacterial colonies were grown in TSB overnight. Cultures were diluted to 0.1, drop plated on SCMA plates and incubated at 37°C for 18-20h. Precipitation zones were measured with a standard metric ruler.

## Statistical analysis

Data were analysed using GraphPad Prism Software (GraphPad Software, Version 10.2.3). For statistical analysis, at least three independent experiments were performed (sample size is indicated for each graph). All data are presented as means with standard deviation (±SD). P-values ≤0.05 were considered significant. When two groups were analysed, a diagnostic t-test was run first. If residuals were normally distributed (Shapiro-Wilk test), samples were analysed with an unpaired, two-tailed t-test. If residuals were not normally distributed, the non-parametric Mann-Whitney test (unpaired, two-tailed) was used instead.

When more than two groups were compared, a diagnostic one-way ANOVA was carried out first to check for normality of residuals (Shapiro-Wilk test), followed by Brown-Forsythe test to check for equal variance. If the ANOVA assumptions were met, the test was followed by an appropriate post-hoc test, as indicated for each graph. The Brown-Forsythe and Welch ANOVA test were used whenever the condition for equal variance was not met. If the ANOVA assumptions were not met, the non-parametric Kruskal-Wallis test, followed by Dunn's multiple comparisons test was applied instead.

A two-way ANOVA was applied for time-course CFU experiments on log10-transformed CFU data, followed by an appropriate post-hoc test, as indicated for each graph.

No matching or pairing was assumed in any ANOVA test.

## Supporting information

**S1 Fig.** *brnQ1* complementation *in trans* restores growth of *S. aureus* JE2 *brnQ1.* A functional copy of the *brnQ1* gene, under its native promoter region, (SAUSA300_0188/ SAUSA300_RS00985) was cloned into a high-copy plasmid and used for complementing JE2 *brnQ1* to yield JE2 *brnQ1*+p*brnQ1*. To exclude potential negative effects on bacterial fitness caused by gene overexpression from a high-copy plasmid, constructs carrying the *brnQ1* gene without promoter sequence were created (termed "$P_{less}$"). **(a)** Effect of the complementation plasmid (p*brnQ1*) or promoter-less *brnQ1* plasmid ($P_{less}$) on bacterial growth in a chemically defined medium with 1 mM of each BCAA (CDM). $OD_{600}$ was determined every 18 min, for 20h. Data are shown as mean values from independent experiments ± SD (n = 3). **(b)** Doubling times ($T_d$) estimated from the growth curve. Dotted line corresponds to mean $T_d$ for JE2 WT. Statistical analysis: one-way ANOVA, with Dunnett's multiple comparisons (vs JE2 WT)*; ns = not significant*; **p < 0.01*.
(TIF)

**S2 Fig. CFU numbers recovered from macrophages at 6h *p.i.* are comparable among the tested *S. aureus* strains.**
**(a)** Bacteria recovered from infected macrophages at 6h *p.i.* a Raw CFU counts (CFU/mL) plotted from data set shown in Fig 1d. Data are shown as mean ±SD from independent experiments (JE2 WT and JE2 *brnQ1*: n = 7, JE2 *brnQ1*+p*brnQ1* and Cowan I: n = 3). **(b)** CFU counts shown in a, normalised to "T0" (i.e., 45 min *p.i.*). Numbers inside bars represent the plotted value (mean). Statistical analysis (**a**): one-way ANOVA, with Dunnett's multiple comparisons, vs JE2 WT; *ns = not significant.*
(TIF)

**S3 Fig. *In vitro* growth of *S. aureus* JE2 *brnQ1* mutant in different culture media.** *S. aureus* growth in **(a)** the nutrient-rich tryptic soy broth medium (TSB), **(b)** RPMI1640 and **(c)** *Infection Medium* (RPMI1640 + 10% v/v heat inactivated FBS). $OD_{600}$ was measured every 18 min, for 20h. Data are shown as mean values ± SD from independent experiments: (a) JE2 WT and JE2 *brnQ1*: n = 5, JE2 *brnQ1*+p*brnQ1*: n = 2; (b) JE2 WT and JE2 *brnQ1*: n = 6, JE2 *brnQ1*+p*brnQ1*: n = 4; (c) JE2 WT and JE2 *brnQ1*: n = 3, JE2 *brnQ1*+p*brnQ1*: n = 1.
(TIF)

**S4 Fig. Phagosomal escape of *S. aureus* JE2 *brnQ1* mutant in epithelial cells. (a)** Dynamics of *S. aureus* infection in professional phagocytes (e.g., macrophages) vs non-professional phagocytes (e.g., epithelial or endothelial cells). Phenol-soluble modulins (*psm*), the agr quorum sensing system (*agr*) and the non-ribosomal peptide synthetase AusAB (*ausAB*) were reported as bacterial factors involved in the phagosomal escape of *S. aureus* (modified after [7]). Live cell imaging of reporter **(b)** HeLa cells or **(c)** human bronchial epithelial cells 16HBE14o- constitutively expressing the phagosomal escape marker YFP-CWT, were infected with mRFP-expressing JE2 WT or JE2 *brnQ1*. Green arrows indicate phagosomal escape events and white arrows indicate intracellular bacterial replication. Shown are micrographs extracted from time-lapse series, representative of 3 independent experiments (n = 3). Scale bars = 10 µm (see also S2–S5 Movies). (*16HBE14o-: immortalized human bronchial epithelial cells*).
(TIF)

**S5 Fig. *S. aureus brnQ1* mutants show delayed phagosomal escape and intracellular replication in non-professional phagocytes. (a)** Experimental procedure for epithelial cell infections: cells were infected with *S. aureus* at a multiplicity of infection (MOI) of 5 for 1h, following synchronization by centrifugation. Extracellular *S. aureus* were removed by a 30 min treatment with 20 µg/µL lysostaphin. Infected cells were further incubated in the presence of 2 µg/mL lysostaphin. "T0" corresponds to 1.5h *p.i.* **(b)** Phagosomal escape rates of *S. aureus* JE2 WT and Nebraska Transposon Mutant Library (NTML) *S. aureus* mutants within several genes associated with either BCAA-uptake (*brnQ1-3* and *bcaP*) or biosynthesis (*ilvE, ilvD, leuA*) were assessed by automated fluorescence microscopy at 4h *p.i.,* in HeLa YFP-CWT phagosomal escape reporter cells. Phagosomal escape rates of the JE2 WT (percent YFP-CWT positive events of total internalized bacteria) were set to 100% (dotted line). *S. aureus* Cowan I was used as phagosomal escape-negative

control. Data are shown as mean±SD from independent experiments (n=2). "NE" designates mutant strain identifier in the NTML. All bacterial strains express the mRFP fluorescent protein. **(c)** Phagosomal escape was quantified in reporter HeLa or 16HBE14o⁻ cells expressing the YFP-CWT phagosomal escape marker, by automated fluorescence microscopy, at different time-points, using SNARF-stained bacteria. Data represent the percent YFP-CWT positive events (*phagosomal escape*) of total internalized bacteria. Data are shown as mean±SD from independent experiments (n>3). **(d)** Intracellular replication in HeLa and 16HBE14o- epithelial cells. Data are shown as means±SD from independent experiments (n>3). **(e)** Cytotoxicity of *S. aureus* infection against HeLa and 16HBE14o- epithelial cells (% LDH release). Data are shown as means±SD from independent experiments (n>3). Statistical analysis: **(c)** two-way ANOVA with Dunnett's multiple comparisons test (each sample vs JE2 WT, for each time-point); **(d)** two-way ANOVA with Dunnett's multiple comparisons test, using log10-transformed data (each sample vs JE2 WT, for each time-point); **(e)** Kruskal-Wallis test with Dunn's multiple comparisons test (JE2 WT vs *brnQ1*, for each time-point); *ns=not significant; \*p<0.05; \*\*p<0.01; \*\*\*p<0.01; \*\*\*\*p<0.0001.* (*16HBE14o-: immortalized human bronchial epithelial cells*). **(a)** contains icons which were modified (colour and design) from https://bioicons.com/, as follows: petri-dish-lid-yellow icon by Servier https://smart.servier.com/ is licensed under CC-BY 3.0 Unported https://creativecommons.org/licenses/by/3.0/.
(TIF)

**S6 Fig. Supernatants of *S. aureus* JE2 *brnQ1* grown in TSB are cytotoxic to HeLa cells and JE2 *brnQ1* exhibits normal haemolysis on sheep blood agar. (a)** Treatment with 10% sterile stationary phase supernatant (24h, TSB) and its effect on cytotoxicity (% LDH release) against HeLa cells (4h of treatment). *S. aureus* Cowan I is not cytotoxic against epithelial cells and serves as control. Data are shown as mean values±SD of independent experiments (JE2 WT and JE2 *brnQ1*: n=4, JE2 p*brnQ1*+p*brnQ1* and Cowan I: n=3). Statistical analysis: one-way ANOVA, with Dunnett's multiple comparisons test, vs JE2 WT; *ns=not significant; #=below detection limit, set to "0".* **(b, c)** *S. aureus* JE2 haemolysis on Columbia agar with 5% defibrinated sheep blood, for 30h, at 37°C: **(b)** JE2 WT and **(c)** JE2 *brnQ1*. Representative image is shown (n>3).
(TIF)

**S7 Fig. Dual RNA-seq of *S. aureus* JE2- infected macrophages. (a)** Principal component analysis (PCA) of host transcripts and **(b)** intracellular bacterial transcripts from dual RNA-seq, from 3 independent experiments (n=3) (*hMDM: primary human monocyte-derived macrophages*).
(TIF)

**S8 Fig. Visualization of major *S. aureus* regulons that are differentially regulated in intracellular *S. aureus* JE2 *brnQ1* compared to JE2 WT.** Displayed squares represent a gene of the indicated regulon and box coloration was assigned according to the heat map in the lower right (red: upregulated in JE2 *brnQ1* compared to JE2 WT; blue: downregulated in JE2 *brnQ1* compared to JE2 WT). Gene-boxes with black frames indicate significance of differential gene expression (padj< 0.05) and absolute values of log2 fold-changes >1.5 or <-1.5. Genes regulated upon bacterial stringent response, such as genes coding for ribosomal proteins or tRNAs, were added to the analysis (framed gene set in the upper right). Gene identifiers correspond to USA300_FPR3757 (old locus ID). Regulons were retrieved from RegPrecise.lbl.gov.
(TIF)

**S9 Fig. The *S. aureus*- containing phagosome is an iron deplete environment.** Transcriptomes of intracellular JE2 WT in primary human macrophages at 10h *p.i.* compared to bacteria grown in *Infection Medium* for 10h. Horizontal lines represent a p-value cutoff<0.05. Vertical lines represent cutoffs of log₂fold changes of either <-1.5 or > 1.5. Experiment was run in parallel with experiments shown in Fig 2a-b. Data from Fig 2b were used for comparison and expressed as "intracellular vs *Infection Medium*" using DESeq2. Transcripts of iron uptake systems, which are prominently differentially regulated in intracellular bacteria vs *Infection Medium,* are depicted in green. Data are shown from 3 independent experiments (n=3).
(TIF)

**S10 Fig.  During growth under iron-replete conditions, *S. aureus* JE2 *fur* and JE2 *brnQ1*/*fur* upregulate iron uptake mechanisms.** Total iron ($Fe^{2+}$ and $Fe^{3+}$) measured in bacteria grown for 24h in RPMI1640 medium supplemented with 20 µM $Fe^{2+}$ (provided as $FeSO_4$). Data are shown as mean values ±SD of independent experiments (n = 2). (TIF)

**S11 Fig.  *Cerulean* is inducible by aTC in TSB medium in *S. aureus* JE2. (a)** JE2 WT and **(b)** JE2 *brnQ1* were grown in TSB medium supplemented with 300 ng/mL aTc (concentration used in infection experiments), 150 ng/mL or vehicle control (100% ethanol). $OD_{600}$ and fluorescence ($Ex/Em_{433/504\ nm}$) were measured every 18 min for 16h. Data are shown as mean values ±SD of independent experiments (n = 3). (AFU = arbitrary fluorescence units, expressed as *Cerulean*/$OD_{600}$ ratio). (TIF)

**S12 Fig.  *Cerulean* signal quantification in intracellular *S. aureus* JE2.** Dot plots detailing the quantification of the percentage of *Cerulean*-expressing bacteria of all bacteria (labelled with Vancomycin BODIPY FL). Individual bacteria were identified by fluorescence microscopy and were labelled as regions -of-interest (ROI). For each ROI, fluorescence mean intensities in *Cerulean* and BODIPY-FL were measured and were plotted to visualize *Cerulean* expression of single bacteria after induction at 1h and 10h *p.i.* (TIF)

**S13 Fig.  Valine and leucine rescue the growth defect of *brnQ1*- deficient *S. aureus* in macrophages. (a)** *S. aureus* JE2 growth in a chemically defined media, either containing 1 mM of each BCAA (CDM) or lacking the stated amino acid(s). $OD_{600}$ was measured every 18 min, for 48h. Data are shown as mean values ±SD of independent experiments (n = 4). **(b)** Intracellular growth of a *S. aureus brnQ1* in RAW 264.7 macrophages. Macrophages were infected with either WT USA300 or *S. aureus brnQ1* and, after gentamicin treatment, supplemented with either valine (Val), leucine (Leu), or Leu and Val (each at 1 mM final concentration) for the duration of the experiment. Data are shown as the mean log10 value ± S.D. for the calculated fold change in CFU/mL at 20h relative to 1.5h *p.i.*, for each bacterial strain. Each data point plotted represents a biological replicate derived from at least three independent experiments (n ≥ 3). Statistical analysis: Brown-Forsythe and Welch ANOVA with Dunnett's T3 multiple comparison test. **(c)** Fluorescence microscopy micrographs of RAW 264.7 macrophages infected with *S. aureus* USA300 WT and *S. aureus brnQ1* mutant expressing GFP (green) that were labeled with a fluorescent proliferation dye (blue). The macrophage plasmalemma and extracellular cocci are stained with TMR-WGA (red). 1 mM Val and Leu (each) were added to the medium after gentamicin treatment (1.5h *p.i.*). At the outset (i.e., 1.5h) all bacteria are GFP and proliferation dye positive; over time and with replication, GFP-positive yet proliferation dye negative bacteria that are devoid of WGA can be seen at 10h and 20h *p.i.* White arrows indicate unrestricted bacterial replication (*URG*). Shown are representative micrographs, from 2 independent experiments (n = 2) (scale bars = ∼ 10 µm). (TIF)

**S14 Fig.  Inactivation of *codY* rescues the growth of *brnQ1*-deficient *S. aureus* in valine-free CDM. (a)** The ability of *S. aureus* USA300 WT, a *codY* mutant (*codY*::Tn), a *brnQ1* mutant, a *brnQ1*/*codY* double mutant (Δ*brnQ1*/*codY*::Tn) and a *brnQ1*/*codY*/*ilvD* triple mutant to grow in Val-free CDM are shown. **(b)** CDM was supplemented with either 1mM Val or **(c)** 0.1% (v/v) tryptone (as a source of peptides) as indicated. Plotted data represent endpoint $OD_{600}$ readings, measured at 20h post-inoculation and the bars represent the mean ± SD for the indicated strains in each condition. Individual data points represent independent biological replicates, from three independent experiments (n = 3). Statistical analysis: Kruskal-Wallis with Dunn's multiple comparisons test performed on the data deriving from each condition (vs USA300 WT); **$p < 0.01$, ns = not significant. (TIF)

**S15 Fig. Unrestricted growth *(URG)* of intracellular *S. aureus brnQ1* mutants during long-term infection of macrophages occurs in a stochastic manner.** Macrophages were generated from different donors (JE2 *brnQ1*: n = 4; 6850 *brnQ1*, SH1000 *brnQ1* and RN4220 *brnQ1*: n = 2) and seeded in well plates as technical replicates (at least 3 wells; JE2 *brnQ1* Donor #1: 72 wells; JE2 Donor #2: 80 wells). Infected plates were observed daily and the number of wells, as well as the time point when *URG* occurred were recorded and plotted in a Kaplan-Meier survival curve. Each well represents in this case a *test subject*. Wells infected with the parental wild-type strain served as positive controls.
(TIF)

**S16 Fig. Protease production is increased in quadruple *brnQ1/bcaP/codY/ilvD* mutants, yet they fail to overcome macrophage restriction. (a)** Quantitation of the zones of precipitation around the bacteria grown on SCMA plates for the detection of proteolytic activity shown in **(b)**. Data are shown as mean values ± SD, from independent experiments (n > 3). Statistical analysis: Kruskal-Wallis test with Dunn's multiple comparisons test; **p < 0.01, ***p < 0.001, ns = not significant. (*codY* = USA300 *codY::Tn*; Δ*brnQ1*/Δ*bcaP* = USA300 double deletion mutant of *brnQ1* and *bcaP*; Δ*brnQ1*/Δ*bcaP*/*codY* = USA300 triple mutant Δ*brnQ1*/Δ*bcaP*/*codY::Tn*; Δ*brnQ1*/Δ*bcaP*/*codY*/*ilvD* = USA300 quadruple mutant Δ*brnQ1*/Δ*bcaP*/*codY::Tn*/*ilvD::Tn-Kan*[R]). Asterisks (magenta) designate strains where *unrestricted growth* (*URG*) in macrophages was observed (data shown in Fig 1c and Fig 4e).
(TIF)

**S1 Movie. Live cell imaging of *S. aureus*-infected primary macrophages. mRFP-expressing *S. aureus* JE2 WT or JE2 *brnQ1* (red). Host cell death (green fluorescence) is detected by CellEvent Caspase-3/7 Green Detection Reagent. Scale bar=100 µm.**
(MP4)

**S2 Movie. Phagosomal escape of *S. aureus* JE2 WT in HeLa cells.** (red: *S. aureus*; yellow signal accumulation or ring-like structures: phagosomal escape marker). Shown is a time-lapse movie, representative of 3 independent experiments (n = 3). White frame highlights a region of interest depicted in S4b Fig. Scale bar = 50 µm.
(MP4)

**S3 Movie. Phagosomal escape of *S. aureus* JE2 *brnQ1* mutants in HeLa cells.** (red: S. *aureus*; yellow signal accumulation or ring-like structures: phagosomal escape marker). Shown is a time-lapse movie, representative of 3 independent experiments (n = 3). White frame highlights a region of interest depicted in S4b Fig. Scale bar = 50 µm.
(MP4)

**S4 Movie. Phagosomal escape of *S. aureus* JE2 WT in bronchial epithelial cells.** Live cell imaging of reporter 16HBE14o- cells constitutively expressing the phagosomal escape marker YFP-CWT, infected with mRFP-expressing *S. aureus* JE2 WT. (red: *S. aureus*; yellow signal accumulation or ring-like structures: phagosomal escape marker). Shown is a time-lapse movie, representative of 3 independent experiments (n = 3). Scale bar = 50 µm.
(MP4)

**S5 Movie. Phagosomal escape of *S. aureus* JE2 *brnQ1* mutants in bronchial epithelial cells.** Live cell imaging of reporter 16HBE14o- cells constitutively expressing the phagosomal escape marker YFP-CWT, infected with mRFP-expressing *S. aureus brnQ1*. (red: *S. aureus*; yellow signal accumulation or ring-like structures: phagosomal escape marker). Shown is a time-lapse movie, representative of 3 independent experiments (n = 3). Scale bar = 50 µm.
(MP4)

**S1 Dataset. Expression profiling by high throughput sequencing.** *S. aureus* JE2 wild-type (WT) or its isogenic JE2 *brnQ1* mutant were either used to infect primary human macrophages for 10h or were cultivated in *Infection Medium* (RPMI1640 + 10% v/v heat-inactivated FBS) for 10h. RNA of both pathogen and host cells were isolated together (dual RNA-Seq of infected macrophages) or isolated from planktonic bacteria (*Infection Medium*) after 10h of infection/cultivation. DESeq2 was used to analyse the following conditions: host transcripts in human macrophages infected with JE2 *brnQ1* compared to infected with JE2 WT (*HUMAN-ic_Mut_vs_ic_WT*); bacterial transcripts for intracellular JE2 *brnQ1* compared to intracellular JE2 WT (*BACT-ic-Mut_vs_ic_WT*); bacterial transcripts in intracellular bacteria compared to *Infection Medium* for JE2 *brnQ1* (*BACT-ic_Mut_vs_RPMI_Mut*) or JE2 WT (*BACT-ic_Mut_vs_RPMI_WT*) (*HUMAN: human transcripts; BACT: bacterial transcripts; Mut: JE2 brnQ1; WT: JE2 WT; ic: intracellular; RPMI: Infection Medium*). (XLSX)

**S1 Table. Bacterial strains used in this study** .
(PDF)

**S2 Table. Plasmids used in this study.**
(PDF)

**S3 Table. Oligonucleotides used in this study.**
(PDF)

**S1 Text. Additional information detailing the role of the BrnQ1 transporter in the interaction of *S. aureus* with epithelial cells (refers to data shown in S4 and S5 Figs and S2-S5 Movies).**
(PDF)

## Acknowledgments

The Leica TCS SP5 CLSM, the Perkin Elmer Operetta High-Content Imaging System and the JEOL JEM-2100 TEM were funded by the Deutsche Forschungsgemeinschaft (DFG, German Research Foundation) under project codes 116162193, 237502929 and 218894163, respectively.

RNA-seq was performed by the Core Unit SysMed at the University of Würzburg and was supported by the IZKF at the University of Würzburg (project Z-6).

We thank Jürgen Fritsch (University Hospital of Regensburg, Department of Infection Prevention and Infectious Diseases) for genome sequencing. We thank Simon Foster (University of Sheffield) for providing the *S. aureus* MJH010 *fur*::tet strain. We thank Claudia Gehrig-Höhn and Daniela Bunsen for support in transmission electron microscopy sample processing. We thank Nadine Vollmuth for valuable scientific input.

## Author contributions

**Conceptualization:** Adriana Moldovan.

**Data curation:** Adriana Moldovan, Ronald S Flannagan, Marcel Rühling, Martin J Fraunholz.

**Formal analysis:** Adriana Moldovan, Ronald S Flannagan, Marcel Rühling.

**Funding acquisition:** Adriana Moldovan, Marcel Rühling, David E Heinrichs, Thomas Rudel, Martin J Fraunholz.

**Investigation:** Adriana Moldovan, Ronald S Flannagan, Marcel Rühling, Kathrin Stelzner, Clara Hans, Kerstin Paprotka, Tobias C. Kunz, Martin J Fraunholz.

**Methodology:** Adriana Moldovan, Ronald S Flannagan, Martin J Fraunholz.

**Project administration:** Adriana Moldovan, David E Heinrichs, Thomas Rudel, Martin J Fraunholz.

**Supervision:** Adriana Moldovan, David E Heinrichs, Thomas Rudel, Martin J Fraunholz.

**Validation:** Adriana Moldovan, Ronald S Flannagan.

**Visualization:** Adriana Moldovan, Martin J Fraunholz.

**Writing – original draft:** Adriana Moldovan.

**Writing – review & editing:** Adriana Moldovan, Ronald S Flannagan, Marcel Rühling, David E Heinrichs, Thomas Rudel, Martin J Fraunholz.

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
