## [Decision Letter · Decision Letter 0]

PPATHOGENS-D-25-00597

Inactivation of branched-chain amino acid uptake halts *Staphylococcus aureus* growth and induces bacterial quiescence within macrophages

PLOS Pathogens

Dear Dr. Fraunholz,

Thank you for submitting your manuscript to PLOS Pathogens. After careful consideration, we feel that it has merit but does not fully meet PLOS Pathogens's publication criteria as it currently stands. Therefore, we invite you to submit a revised version of the manuscript that addresses the points raised during the review process.

Please submit your revised manuscript within 60 days Jun 02 2025 11:59PM. If you will need more time than this to complete your revisions, please reply to this message or contact the journal office at plospathogens@plos.org. Please include the following items when submitting your revised manuscript:

We look forward to receiving your revised manuscript.

Kind regards,

Alice Prince

Section Editor

PLOS Pathogens

Alice Prince

Section Editor

PLOS Pathogens

 Sumita Bhaduri-McIntosh

Editor-in-Chief

PLOS Pathogens

orcid.org/0000-0003-2946-9497

 Michael Malim

Editor-in-Chief

PLOS Pathogens

orcid.org/0000-0002-7699-2064

**Journal Requirements:**

- ® on pages: 25, 26, and 28.

Potential Copyright Issues:

- Figures 1, 3, and 5. Please confirm whether you drew the images / clip-art within the figure panels by hand. If you did not draw the images, please provide (a) a link to the source of the images or icons and their license / terms of use; or (b) written permission from the copyright holder to publish the images or icons under our CC BY 4.0 license. Alternatively, you may replace the images with open source alternatives. See these open source resources you may use to replace images / clip-art:

6) Please ensure that the funders and grant numbers match between the Financial Disclosure field and the Funding Information tab in your submission form. Note that the funders must be provided in the same order in both places as well.

**Reviewers' Comments:**

Reviewer's Responses to Questions

**Part I - Summary**

Reviewer #1: The study by Moldovan et al describes the consequences of a mutation in the branched chain amino acid transporter BrnQ1 for intracellular growth and protein synthesis in S. aureus. Interestingly, mutation of codY restores growth, showing that repression of BCAA biosynthesis genes occurs in the macrophage, although the mechanism responsible for this is unexplored. Aside from the key finding (S. aureus uses exogenous BCAAs in the phagosome and requires BrnQ1), the rest of the paper is of more limited significance.

Reviewer #2: This manuscript describes the role of branched-chain amino acids (BCAA) in the survival of S. aureus within phagolysosomes after engulfment by macrophages. The authors analyzed the ability of individual S. aureus mutants deficient for the BCAA transporters BrnQ1, BrnQ2, BrnQ3 and BcaP to replicate and survive within macrophages. They also included mutants lacking the ilvE, ilvD and leuA genes involved in BCAA biosynthesis. Of all these tested, only the brnQ1 mutant was found to be defective in intracellular replication within macrophages. Importantly, this phenotype was observed in multiple S. aureus strains and a growth delay was observed in defined medium with low BCAA availability. The brnQ1 mutant also failed to induce death of the macrophages and remained intracellular, unlike other strains, which ultimately escaped and infected neighboring cells.

Transcriptome analysis revealed the downregulation of genes involved in iron acquisition in the brnQ1 mutant compared to the WT strain, along with several other virulence genes, consistent with studies demonstrating the limited iron availability of a phagolysosome. The results here are clear but without more information about the rationale for picking 10 hrs post infection to collect samples for these experiments, it is difficult to completely interpret these results. An interesting aspect of this study was that an iron regulated gene was not expressed in the brnQ1/fur background, suggesting the cells are entering a state of quiescence. This was nicely tested by attempting to induce a reporter gene in the brnQ1 mutant, which demonstrated reduced reporter expression compared with WT cells. In addition, using a combination of mutations that inactivate various genes involved in BCAA regulation, uptake, and/or biosynthesis, the authors conclusively demonstrated that S. aureus could overcome the limited BCAA levels in a phagolysosome by a combination of increased expression of CodY-regulated BCAA transporters or biosynthetic genes. Finally, long-term intracellular survival assays demonstrated the brnQ1 mutant persisted for up to 28 days within a macrophage. These survivors were found to be distinct from classical persisters (SCVs), which exhibit slow growth, antibiotic tolerance, and loss of pigmentation, none of which are characteristics of the brnQ1 mutant cells.

Reviewer #3: This is an interesting manuscript with new insights into the intracellular lifestyle of S. aureus in macrophages depending on BCAA availability.

The manuscript “Inactivation of branched-chain amino acid uptake halts Staphylococcus aureus growth and induces bacterial quiescence within macrophages” by Moldovan et al. investigates the impact of branched-amino acids and the respective transporters of S. aureus on its intracellular fate in macrophages. To show this the authors used macrophages that they developed from monocytes, which they infected with WT S. aureus and different transporter mutants in different S. aureus backgrounds, the complemented mutants and BCAA-synthesis mutants; they performed growth studies with and without external BCAAs; they performed dual RNAseq of infected macrophages revealing that especially iron-dependent genes were differentially expressed in WT and the mutant indicating that brnQ1-mutants are impaired in gene expression, in de novo protein synthesis representing a quiescent status. However, by external BCAAs supplementation the mutant acted comparable to the WT. The authors investigated the role of CodY for the effects observed in the brnQ1-strain by constructing a brnQ1-codY double mutant, which showed intracellular replication and escape of the phagolysosome. The authors also determined the role of CodY as the regulator of BCAA transporters and synthesis genes including the other important BCAA transporter BcaP and the most important synthesis gene ilvD. The authors also performed longterm infection assays revealing that the mutants could survive during this period. The authors tried to show that the persisting mutants were neither classical persisters nor SCVs.

**Part II – Major Issues: Key Experiments Required for Acceptance**

Reviewer #1: The antibiotic killing experiment is unusual in that it does not include a wild-type strain and the level of killing by rifampicin is not compared with anything. These experiments should be done more completely with the wild-type, mutant and multiple antibiotics, or (as I strongly recommend), removed from the manuscript altogether.

Reviewer #2: None

Reviewer #3: Major points of criticism

1. While the authors describe in the introduction the role of S. aureus and also of branched-chain amino acids, it would be interesting for the reader why the authors got the idea that BCAA could play any role for internalized S. aureus.

2. Also, is this an observation which holds only true for professional phagocytes or is this also a mechanism for intracellular survival in non-professional phagocytes such as epithelial or endothelial cells?

3. Since the authors do not only use BCAA transporter mutants, but also mutants in genes for BCAA synthesis, they should introduce the synthesis genes in the introduction. Furthermore, is it known if there is any phenotype associated with the single BCAA synthesis mutants? Can this be expected, since mostly S. aureus uses BCAAs from the environment as it is described in several studies (e.g. Kaiser et al. PlosGen2018), where it also has been described that it has been suggested that S. aureus would be auxotrophic for leucine and valine.

4. Does it really make sense to test a codY mutant in this regard? Since codY is a negative regulator of the BCAA-synthesis genes, in a codY mutant, these genes would be repressed, thereby acting if at all more in a BCAA synthesis way. Therefore, what should be expected in terms of determining the role of the BCAA-synthesis operon?

5. The term of “unrestricted growth” after escape from the macrophage, is this a term used in the scientific community or is it a term only used in this study? The authors should describe the effect of this more intensely.

6. By first mentioning the Nebraska Transposon Library, the authors should state in which background the library has been established, USA300 LAC.

7. Why did the authors use JE2 as another strain for introducing the brnQ1 mutation as indicated line 124? And surprisingly, in line 133 the authors report that the mutation was also moved into the S. aureus strains 6850, SH1000 and RN4220. They should have indicated this earlier.

8. The authors show that the brnQ1 mutation does not only effect intracellular replication but also growth in cell culture and infection media. This is not entirely new as Kaiser et al have already shown in 2015 that this mutant has a growth defect in chemical defined medium without BCAAs (Kaiser et al in 2015, Inf Immun).

9. The paragraph showing that these mutants are not “classical persisters” is not very convincing. The authors do not show data of the real persisters after 28d, but point at the transcriptomic data at 10h p.i. Also, for the antibiotic tolerance, they test infection after 10h. It would be more convincing to test antibiotic tolerance after long-term infection.

However, convincing is the description that the mutants do not undergo the SCV status during persistence.

A last point of interest would be to search the publicly available S. aureus genome data bases to check if there are any natural brnQ1 existing mutants in clinical isolates present. This would add a clinical role for the interesting mutant.

**Part III – Minor Issues: Editorial and Data Presentation Modifications**

Reviewer #1: The longterm survival of the mutant is discussed as though it could represent a long-term survival niche, but the mutant doesn't appear to occur naturally, so this behavior is artifactual and of limited significance. The authors should temper their language when discussing this part of the paper. The comparison with SCVs is of limited relevance as they are clearly completely different things and one would not expect them to be similar. A SCV is generally a S. aureus strain that has lost the ability to respire, while this mutant has lost the ability to take up exogenous BCAAs. Comparison of the two seems unusual. It is recommended that this is reduced or removed or at least justified more thoroughly.

In lines 135-145 the authors discuss different media and their BCAA content but they only give actual concentrations of BCAAs for some conditions and are vague for others. Please add quantities for each condition if they are so relevant.

In line 236, recommend removing "Remarkably" as there is no reason to suspect the Fur repressor would be involved in the phenotype.

Fig. 5e and f should be in supplement.

Reviewer #2: Overall, this is a nicely presented body of work that provides important new insight into the survival and growth of S. aureus after engulfment by macrophages. Although this a very thorough study that addresses numerous aspects of growth within a phagolysosome, my one concern is the characterization of the phenotype observed as “dormancy”, akin to persisters. I agree that it is tempting to speculate that this is a new and previously uncharacterized metabolic state, but without any evidence that this state arises in the WT strain or during an infection, I believe any speculation as to the functional relevance of this state should be made with caution.

Questions and comments:

Line 42: Change, “It is facultative intracellular…”, to “It is a facultative intracellular pathogen…”.

Line 256: In this context, what does deplete mean? Clearly there is enough BCAAs for growth, as long as the bacteria can transport them in.

Line 263: I don’t understand this statement: “During growth in CDM lacking all three BCAAs, the growth of S. aureus is preceded by an extended lag phase of ~20h. Omission of Val and Leu, however, completely halted bacterial growth.” If the absence of all three BCAAs causes a lag in growth, why would the absence of only two “completely halt growth”? Please clarify.

Reviewer #3: Minor points of criticism

Line 58: The description “commensal” should be avoided, see the viewpoint article by Proctor, RA: Clin Infect Dis 2021 Jul 1;73(1):e267-e269. doi: 10.1093/cid/ciaa1431.

Have We Outlived the Concept of Commensalism for Staphylococcus aureus?

A legend to Table 1 is missing.

Figure 1a: This figure can be deleted, the assay is well described and not especially new.

Fig. 1f: What do the authors mean by “stills”?

Fig. 1g: In the legend of this figure there is no mentioning of the mutant. What do the authors intend to show here?

Fig. 3c can be deleted, see commentary to Fig. 1a

Reference list: In the reference list, all bacteria and gene names should be written in italics. Also, the references should be in a homogenous style with either all journals abbreviated or not.

Lines 257ff: The authors suggest that they would test the effects of BCAAs to intracellular bacterial growth directly, which is fine. However, next the report at first the results of the effect on isdB expression. It would be more logic to show at first the results of the effects of external BCAAs on growth and then on isdB expression.

Supporting Fig. 12: The authors should also test the effects of “no valine” on a single codY-mutants since the growth of the WT is also affected in this medium.

Materials and Methods

Line 617: Usually, the Thoma counting chamber is used to count eukaryotic cells. How should this work to count bacteria? This should be explained in more detail.

Line 626: This sentence should be corrected “where” to “with”? Also, “mins” should be corrected to “min”.

Also, the authors should use the same abbreviations throughout the manuscript: hours or h with or without a free space.

PLOS authors have the option to publish the peer review history of their article (what does this mean? ). If published, this will include your full peer review and any attached files.

**Do you want your identity to be public for this peer review?** For information about this choice, including consent withdrawal, please see our Privacy Policy .

Reviewer #1: No

Reviewer #2: No

Reviewer #3: **Yes: ** Barbara C. Kahl

**Figure resubmission:**
---

## [Editor Report · Decision Letter 1]

Dear Dr. Fraunholz,

We are pleased to inform you that your manuscript 'Inactivation of branched-chain amino acid uptake halts Staphylococcus aureus growth and induces bacterial quiescence within macrophages' has been provisionally accepted for publication in PLOS Pathogens.

Best regards,

Alice Prince

Section Editor

PLOS Pathogens

Alice Prince

Section Editor

PLOS Pathogens

Sumita Bhaduri-McIntosh

Editor-in-Chief

PLOS Pathogens

orcid.org/0000-0003-2946-9497

Michael Malim

Editor-in-Chief

PLOS Pathogens

orcid.org/0000-0002-7699-2064

The authors have done an excellent job responding to the reviewers' suggestions. This is a nice contribution.
---

## [Editor Report · Acceptance letter]

Dear Dr. Fraunholz,

We are delighted to inform you that your manuscript, "Inactivation of branched-chain amino acid uptake halts Staphylococcus aureus growth and induces bacterial quiescence within macrophages," has been formally accepted for publication in PLOS Pathogens.

Best regards,

Sumita Bhaduri-McIntosh

Editor-in-Chief

PLOS Pathogens

orcid.org/0000-0003-2946-9497

Michael Malim

Editor-in-Chief

PLOS Pathogens

orcid.org/0000-0002-7699-2064